# MambaVoiceCloning: Efficient and Expressive Text-to-Speech via State-Space Modeling and Diffusion Control

**Sahil Kumar**
PhD Program in Mathematics
Yeshiva University
New York, NY 10033, USA
`skumar4@mail.yu.edu`

**Namrataben Patel**
PhD Program in Mathematics
Yeshiva University
New York, NY 10033, USA
`npatel13@mail.yu.edu`

**Honggang Wang**
Department of Computer Science & Engineering
Yeshiva University
New York, NY 10033, USA
`honggang.wang@yu.edu`

**Youshan Zhang**[*]
School of Artificial Intelligence
Chuzhou University
Anhui, 239000, China
`youshan_zhang@chzu.edu.cn`

## Abstract

MambaVoiceCloning (MVC) asks whether the conditioning path of diffusion-based TTS can be made fully SSM-only at inference—removing all attention and explicit RNN-style recurrence layers across text, rhythm, and prosody—while preserving or improving quality under controlled conditions. MVC combines a gated bidirectional Mamba text encoder, a Temporal Bi-Mamba supervised by a lightweight alignment teacher discarded after training, and an Expressive Mamba with AdaLN modulation, yielding linear-time $\mathcal{O}(T)$ conditioning with bounded activation memory and practical finite look-ahead streaming. Unlike prior Mamba–TTS systems that remain hybrid at inference, MVC removes attention-based duration and style modules under a fixed StyleTTS2 mel–diffusion–vocoder backbone. Trained on LJSpeech/LibriTTS and evaluated on VCTK, CSS10 (ES/DE/FR), and long-form Gutenberg passages, MVC achieves modest but statistically reliable gains over StyleTTS2, VITS, and Mamba–attention hybrids in MOS/CMOS, $F_0$ RMSE, MCD, and WER, while reducing encoder parameters to 21M and improving throughput by $1.6\times$. Diffusion remains the dominant latency source, but SSM-only conditioning improves memory footprint, stability, and deployability. Code: `https://github.com/sahilkumar15/MVC`.

## 1 Introduction

Text-to-Speech (TTS) systems continue to improve in naturalness and expressive control Li et al. (2023b); Kim et al. (2021); Ning et al. (2019); Tan et al. (2021), yet most conditioning stacks rely on transformer attention Vaswani et al. (2023); Wang et al. (2017) or recurrent modules. Attention introduces quadratic computational and memory complexity and global context mixing, while recurrent architectures exhibit long-range drift and unstable memory dynamics. Linear attention variants Choromanski et al. (2021) reduce asymptotic cost but preserve global interactions that complicate streaming. Meanwhile, diffusion decoders Popov et al. (2021); Huang et al. (2022); Liu et al. (2022); Kong et al. (2020); Zhang et al. (2023) dominate inference runtime, making encoder efficiency central to deployment.

**Why Mamba vs. Transformer/RNN.** State-space models (SSMs), particularly Mamba Gu & Dao (2024), provide bounded activations, linear-time sequence scans, and state-persistent streaming. These properties reduce memory pressure relative to attention-based models and mitigate drift in

---

[*]Corresponding author. This research was funded by the Research Project of Chuzhou University (Grant No. 2025qd36).

recurrent architectures, supporting stable conditioning over multi-sentence inputs. However, existing Mamba–TTS systems Jiang et al. (2024); Zhang et al. (2024) remain hybrid at inference, retaining attention-based duration or style modules that limit streaming robustness.

This work investigates whether diffusion-based TTS can adopt a fully SSM-only conditioning stack at inference for text, rhythm, and prosody under a strictly matched mel–diffusion–vocoder pipeline. The StyleTTS2 decoder and vocoder remain fixed; only the conditioning path is redesigned. MVC introduces three selective SSM modules: a gated bidirectional Mamba text encoder; a Temporal Bi-Mamba aligned using a lightweight monotonic teacher during training only; and an Expressive Mamba with AdaLN modulation. A gated forward–backward fusion mechanism replaces concat-only bi-Mamba fusion used in prior work.

**Why NaturalSpeech 3, CosyVoice 3, and HiggsAudio-V2 are not direct baselines.** Industrial-scale TTS systems such as NaturalSpeech 3 Ju et al. (2024), CosyVoice 3 Du et al. (2025), and HiggsAudio-V2 Boson AI (2025) rely on multi-hundred-thousand– to million-hour proprietary multilingual corpora, LLM-scale semantic encoders, and multi-stage pipelines. Their performance is driven primarily by scale rather than conditioning-architecture design, making them unsuitable as decoder-matched baselines for a controlled architectural study. For fairness, comparisons are restricted to open-data systems trained under identical preprocessing, mel front-end, vocoder, and optimization schedules; a detailed contextual comparison is provided in Appendix F.

**Scope of evaluation.** The evaluation covers in-distribution speech (LJSpeech Ito & Johnson (2017), LibriTTS Zen et al. (2019)), zero-shot speakers (VCTK Veaux et al. (2017)), cross-lingual CSS10 (ES/DE/FR) Park & Mulc (2019), and 2–6 minute Gutenberg passages for long-form testing. MVC yields consistent improvements over StyleTTS2 Li et al. (2023b), VITS Kim et al. (2021), and capacity-matched Mamba hybrids Jiang et al. (2024); Zhang et al. (2024) in MOS, CMOS, $F_0$ RMSE, MCD, and WER, while reducing encoder parameters to 21M and improving throughput by a factor of 1.6. Streaming with a finite look-ahead of 0.5–2.0 seconds preserves non-streaming quality, consistent with prior monotonic and streaming sequence modeling approaches, and the diffusion decoder remains the primary latency source Popov et al. (2021); Jeong et al. (2021). Additional runtime, memory, and SSM-sensitivity analyses appear in Appendix A.1.

**Contributions.** (1) A diffusion-based TTS system with a fully SSM-only inference-time conditioning path spanning text, rhythm, and prosody under a fixed decoder. (2) A gated bidirectional Mamba fusion with AdaLN that improves long-range prosody stability and reduces drift on multi-sentence and out-of-distribution text. (3) Protocol- and capacity-matched baselines that isolate the architectural impact of removing inference-time attention. (4) A deployment-oriented analysis covering memory usage, throughput, SSM hyperparameter sensitivity, long-form behavior, and finite look-ahead streaming, demonstrating predictable linear-time characteristics.

## 2 RELATED WORK

TTS conditioning spans attention-based encoders, diffusion decoders, zero-shot systems, and recent state-space models. MVC examines how these paradigms affect efficiency, memory usage, and long-form stability.

**Attention-based TTS.** Transformer-based pipelines such as Tacotron, Tacotron2, JETS, StyleTTS, and StyleTTS2 Wang et al. (2017); Shen et al. (2018); Lim et al. (2022); Li et al. (2023b); Vaswani et al. (2023) provide strong alignment and style modeling but rely on quadratic attention complexity. Even linear attention variants Wang et al. (2020); Choromanski et al. (2021) maintain global interactions that couple text, duration, and prosody, making streaming synthesis sensitive to memory usage. These limitations motivate conditioning stacks with linear-time behavior and bounded activations.

**Zero-shot and large-scale systems.** NaturalSpeech 3 Ju et al. (2024), CosyVoice 3 Du et al. (2025), and HiggsAudio-V2 Boson AI (2025) leverage multi-hundred-thousand– to million-hour proprietary corpora, multilingual pipelines, and LLM-scale encoders integrating text understanding and expressive control. Their performance is primarily driven by scale rather than conditioning design and cannot be reproduced under academic budgets or open-data constraints. MVC instead evaluates conditioning architecture under a fixed mel–diffusion–vocoder pipeline and controlled data scale. A contextual comparison appears in Appendix F.

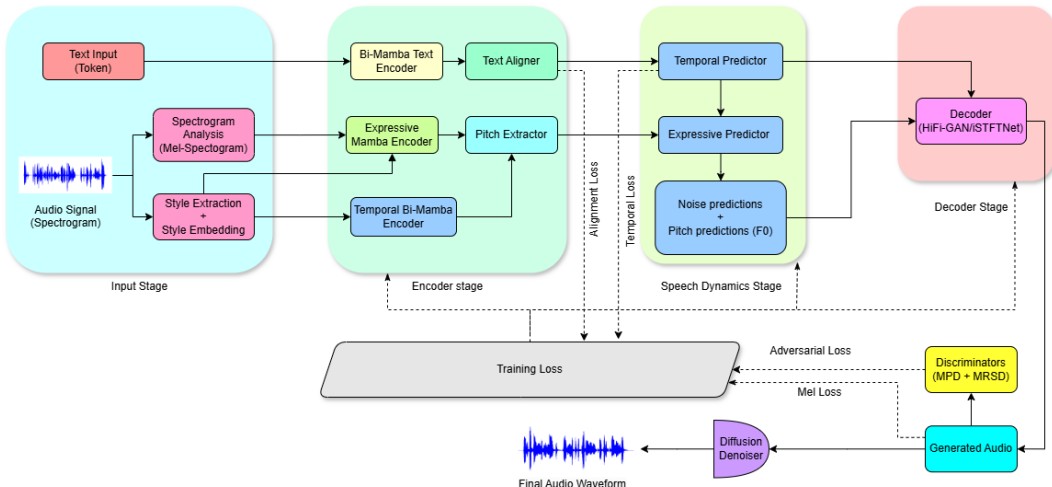

Figure 1: Overview of MambaVoiceCloning (MVC). The framework uses Bi-Mamba Text Encoders for phoneme modeling, a Temporal Bi-Mamba for rhythmic alignment, and an Expressive Mamba for prosodic control. A lightweight aligner (dotted box) provides phoneme–frame supervision only during training, ensuring an SSM-only encoder at inference. Conditioning features drive a diffusion decoder and vocoder for waveform synthesis.

**SSMs and Mamba hybrids.** Mamba introduces input-gated selective scans for linear-time modeling with bounded activations Gu & Dao (2024). SSMs have been explored in speech enhancement, ASR, and hybrid TTS encoders Miyazaki et al. (2024); Jiang et al. (2024); Kumar et al. (2024); Zhang et al. (2024). However, existing Mamba–TTS systems remain hybrid at inference, reintroducing attention or recurrence for duration and style modeling and limiting streaming stability. Runtime and finite look-ahead behavior under diffusion decoders are rarely analyzed.

**Positioning of MVC.** MVC eliminates attention and recurrence across the entire inference-time conditioning stack for text, rhythm, and prosody, retaining a lightweight aligner only during training. It replaces concat-only bi-Mamba fusion Jiang et al. (2024); Zhang et al. (2024) with gated forward–backward fusion and AdaLN modulation. Protocol- and capacity-matched baselines isolate the architectural effects of removing inference-time attention and introducing gated AdaLN fusion (Table 12, Appendix A.2).

## 3  METHODOLOGY

MVC replaces all inference-time attention and recurrence with selective state-space models (SSMs) for text, rhythm/duration, and prosody. A lightweight attention-based aligner provides phoneme–frame supervision during training and is discarded at inference. This yields an SSM-only conditioning stack with linear-time scans and bounded activations. Unlike prior bi-Mamba encoders Jiang et al. (2024); Zhang et al. (2024), MVC employs a gated bidirectional Mamba text encoder, a Temporal Bi-Mamba, and an Expressive Mamba with AdaLN conditioning. All Mamba blocks use a state dimension of 96, depthwise convolution kernel size 5, and gating temperature $\tau = 1.0$ unless otherwise specified.

**High-level overview.** Figure 1 summarizes MVC. From phonemized text and reference audio, MVC produces three conditioning streams: a gated Bi-Mamba text encoder, a Temporal Bi-Mamba for rhythm/duration, and an Expressive Mamba operating on mel spectrograms with AdaLN. These are fused in a speech-dynamics stage and passed to the fixed StyleTTS2 decoder and vocoder. Because decoder and vocoder components are identical across MVC and all baselines (StyleTTS2, VITS, JETS, Hybrid-Mamba, Bi-Mamba), differences in MOS/CMOS, WER, pitch stability, and runtime directly reflect conditioning-stack design. During training, the aligner provides soft phoneme–frame weights; at inference it is discarded, and all encoder modules run in $\mathcal{O}(T)$ without attention maps. For streaming, the bidirectional text encoder is replaced by a causal Uni-Mamba with look-ahead $L$ (Sec 5.3), enabling explicit latency–context trade-offs.

**Notation.** Let $T_x$ and $T_m$ denote the number of text tokens and mel frames, respectively; $d$ the text embedding dimension; $d_h$ the SSM hidden dimension; and $d_s$ the style-embedding dimension. We write $\mathbf{x} \in \mathbb{R}^{T_x \times d}$ for token embeddings, $\mathbf{M} \in \mathbb{R}^{F \times T_m}$ for log-mel spectrograms, and $\mathbf{e} \in \mathbb{R}^{d_s}$ for the global style vector. A compact symbol table in Appendix B.1 consolidates notation and abbreviations for readability.

## 3.1 INPUT PROCESSING

Given waveform $\mathbf{s}_{\text{wav}} \in \mathbb{R}^T$ at 24 kHz, we compute an 80-bin log-mel spectrogram $\mathbf{M} \in \mathbb{R}^{F \times T_m}$ using a Hann-window STFT (FFT 1024, hop 256), mel filterbank projection, and log compression with $\epsilon = 10^{-5}$; the full formulation is in Appendix B.2. Text is normalized and phonemized using `phonemizer` Bernard & Titeux (2021), yielding tokens $[w_1, \ldots, w_{T_x}]$ (with language tags for CSS10 ES/DE/FR); phoneme–grapheme and phoneme-level conditioning have been shown to improve prosody and robustness in neural TTS Jia et al. (2021); Li et al. (2023a). Token embeddings and the global style embedding are computed as

$$\mathbf{x} = \text{Embed}([w_1, \ldots, w_{T_x}]) \in \mathbb{R}^{T_x \times d}, \qquad \mathbf{e} = \frac{1}{T_m} \sum_{t=1}^{T_m} f_\theta(\mathbf{M}_{:,t}) \in \mathbb{R}^{d_s}. \tag{1}$$

where $f_\theta$ is a shallow conv/GRU module shared across encoders. This embedding captures coarse timbre and expressiveness and provides a shared conditioning signal, important for long-form stability, zero-shot speakers, and cross-lingual tests (Sec. 5).

## 3.2 ENCODER STACK

The encoder stack contains three SSM modules: (i) a gated Bi-Mamba text encoder (Sec. 3.2.1); (ii) an Expressive Mamba encoder (Sec. 3.2.2); and (iii) a Temporal Bi-Mamba encoder (Sec. 3.2.3). Appendix E.3 and Table 19 shows that moderate hyperparameter variations produce only small changes in MOS and RTF, confirming that performance gains arise from architecture rather than tuning.

### 3.2.1 BI-MAMBA TEXT ENCODER

We replace self-attention with bidirectional Mamba blocks to obtain a linear-time text encoder with bounded activations. Given $\mathbf{x} \in \mathbb{R}^{T_x \times d}$, we project to $d_h$ and apply forward and backward Uni-Mamba scans,

$$\mathbf{h}_f = \text{Mamba}_f(\mathbf{x}), \quad \mathbf{h}_b = \text{Mamba}_b(\mathbf{x}), \tag{2}$$

where each block follows the selective state-space update Gu & Dao (2024), providing $O(T_x)$ complexity and numerically stable recurrent dynamics (Appendix B.3). The linear-time scanning and bounded activation updates ensure that the encoder remains stable on long phoneme sequences, avoiding attention-fragmentation and activation drift that occur in attention-based duration and prosody predictors. These properties are essential for MVC's long-form behavior: forward/backward scans preserve consistent state magnitudes across multi-sentence and multi-minute segments, providing predictable accumulation of prosodic cues without degradation over time.

Prior bi-Mamba TTS encoders combine directions via simple concatenation; MVC instead employs a gated fusion mechanism:

$$\mathbf{h}_T = \big(\sigma(W_g[\mathbf{h}_f; \mathbf{h}_b]) \odot [\mathbf{h}_f; \mathbf{h}_b]\big)W_o, \tag{3}$$

with $W_g \in \mathbb{R}^{2d_h \times 2d_h}$ and $W_o \in \mathbb{R}^{2d_h \times d_h}$. The gating module modulates forward/backward contexts based on local syntactic cues, improving long-range prosody, reducing drift, and maintaining temporal coherence in extended passages (Sec. 5; Tables 2, 3). Appendix E.1 reports gate statistics on 2–6 minute Gutenberg passages, demonstrating that the gating pattern remains stable and does not collapse, thereby confirming the model's robustness under long-form and streaming conditions.

To incorporate speaker/style information, we apply AdaLN using embedding $\mathbf{e}$:

$$\mathbf{h}_{T,s} = \text{AdaLN}(\mathbf{h}_T, \mathbf{e}), \tag{4}$$

where $\text{AdaLN}(\mathbf{z}, \mathbf{e}) = \gamma(\mathbf{e}) \odot \text{LN}(\mathbf{z}) + \beta(\mathbf{e})$. This gated bi-Mamba + AdaLN architecture is not present in prior Mamba–TTS systems; Table 8 shows that removing either mechanism significantly degrades long-form MOS and pitch RMSE.

### 3.2.2 EXPRESSIVE MAMBA ENCODER

The Expressive Mamba encoder injects speaker-specific prosody into the acoustic representation in linear time. Given mel features $\mathbf{M}$ and style embedding $\mathbf{e}$, we apply a gated transformation with AdaLN conditioning (Appendix B.4), followed by a Mamba block:

$$\mathbf{h}_E = \text{Mamba}(\mathbf{h}_{M,s}) \in \mathbb{R}^{T_m \times d_h}, \tag{5}$$

where $\mathbf{h}_{M,s}$ is the style-conditioned input. This module is fully SSM-based (no attention) and captures slow prosodic dynamics over long inputs; removing it produces the largest CMOS drop among encoder components on OOD data (Table 6).

### 3.2.3 TEMPORAL BI-MAMBA ENCODER

The Temporal Bi-Mamba encoder models rhythmic structure and phoneme–duration alignment. The style embedding $\mathbf{e}$ is broadcast over frames and modulated via a shallow gated transform, producing $\mathbf{h}_S \in \mathbb{R}^{T_m \times d_h}$. Forward and backward Mamba blocks plus a local Conv1D then capture context-dependent timing patterns, and their outputs are fused linearly:

$$\mathbf{h}_B = [\mathbf{h}_f; \mathbf{h}_b] \, \mathbf{W}_f. \tag{6}$$

We keep this fusion linear (no second gating) because prosody disentanglement is handled upstream by the text and expressive encoders; Appendix E.3 shows that adding gating here increases activation memory without consistent MOS gains, clarifying why MVC does not use gating in this module.

### 3.3 ALIGNMENT AND PITCH MODELING

**Training-time aligner.** The aligner is a 2-layer transformer with 4 attention heads and hidden dimension 256, trained with a monotonic alignment loss. It maps token encodings $\mathbf{h}_{T,s}$ to frame-level weights $\boldsymbol{\alpha} \in \mathbb{R}^{T_m \times T_x}$. During training only, a lightweight attention-based aligner maps token-level encodings $\mathbf{h}_{T,s}$ to frame-synchronous representations. Given $\mathbf{M}$ and $\mathbf{h}_{T,s}$, the aligner computes attention weights $\boldsymbol{\alpha} \in \mathbb{R}^{T_m \times T_x}$ and an aligned encoding

$$\mathbf{h}_A = \boldsymbol{\alpha} \, \mathbf{h}_{T,s}. \tag{7}$$

The aligner is a 2-layer, 4-head transformer (hidden size 256) used only as a training-time teacher and completely removed at inference. Appendix B.7 perturbs its attention maps and shows that MVC tolerates moderate alignment noise (WER increase $< 0.4$ points, MOS drop $< 0.05$), indicating that MVC does not rely on a perfectly specified aligner and preserving the SSM-only deployment claim.

**Pitch modeling.** Pitch modeling uses both expressive and temporal encodings. We fuse $\mathbf{h}_E$ and $\mathbf{h}_B$ via a gated block to obtain $\mathbf{h}_P \in \mathbb{R}^{T_m \times d_h}$, and predict the final $F_0$ contour via

$$F_0 = \mathbf{h}_P \, \mathbf{W}_F + b_F. \tag{8}$$

This design avoids an additional attention-based pitch predictor; the prosody path remains SSM-only at inference, which is important for bounded-memory streaming.

### 3.4 SPEECH DYNAMICS AND DECODER CONDITIONING

The speech-dynamics stage refines phonetic and prosodic representations into decoder-ready features. Starting from $\mathbf{h}_A$ and $\mathbf{h}_P$, a temporal predictor (Conv1D + SSM) produces a rhythm-aware representation, which is fused with $\mathbf{h}_P$ via a gated block and projected to a fundamental-frequency trajectory $\hat{F}_0$ and residual noise vector $\mathbf{n}$. The final conditioning sequence is

$$\mathbf{h}_D = [\, \hat{F}_0 \, ; \, \mathbf{n} \,] \in \mathbb{R}^{T_m \times (1 + d_h)}, \tag{9}$$

and is passed to the diffusion decoder. All dynamics and fusion operations here use SSMs and point-wise gates, so the conditioning path remains linear-time and attention-free at inference. Additional architectural details appear in Appendix B.4.

## 3.5 DECODER STAGE AND LOSSES

The decoder uses the StyleTTS2 diffusion model Li et al. (2023b); Popov et al. (2021); Kong et al. (2020) with a matched vocoder; MVC modifies only the conditioning path. Given $\mathbf{h}_D$, the decoder predicts $\hat{\mathbf{M}} = \mathrm{DiffusionDecoder}(\mathbf{h}_D; \{\alpha_t\})$, which the vocoder converts to waveform $\hat{\mathbf{s}} = \mathrm{Vocoder}(\hat{\mathbf{M}})$. We reuse the StyleTTS2 multi-period and multi-resolution discriminators (MPD+MRSD) and mel reconstruction loss. The total loss combines mel, adversarial, and alignment terms:

$$\mathcal{L}_{\mathrm{total}} = \lambda_{\mathrm{mel}}\mathcal{L}_{\mathrm{mel}} + \lambda_{\mathrm{adv}}\mathcal{L}_{\mathrm{adv}} + \lambda_{\mathrm{align}}\mathcal{L}_{\mathrm{align}}. \tag{10}$$

Reusing the StyleTTS2 diffusion and vocoder stack ensures protocol-matched comparisons: Tables 4 and 12 show that MVC improves quality, long-form robustness, and encoder efficiency under an identical decoder/vocoder configuration. Full loss definitions appear in Appendix B.5.

**Training procedure and baselines.** MVC is trained on triples $(\mathbf{x}, \mathbf{M}, \mathbf{s}_{\mathrm{wav}})$ using $\mathcal{L}_{\mathrm{total}}$ with AdamW, cosine decay, gradient clipping, EMA, and automatic mixed precision. All hyperparameters (batch size, training steps, SSM configuration) are held fixed across MVC and Mamba-based baselines to ensure protocol-level parity. Full training steps (Algorithm 1) and implementation details for Hybrid-Mamba and Bi-Mamba (Concat-only) are provided in Appendix B.8 and Appendix B.6, where Table 12 shows that MVC's gains persist under strict reproduction controls.

## 4 EXPERIMENTS

## 4.1 DATASETS AND PREPROCESSING

We train on LJSpeech Ito & Johnson (2017) (24 h, 1 spk.) and LibriTTS Zen et al. (2019) (245 h, 1,151 spk.), and evaluate on VCTK Veaux et al. (2017) (109 spk.; zero-shot) and **CSS10** ES/DE/FR Park & Mulc (2019). Audio is resampled to 24 kHz and converted to 80-bin log-mels; text is normalized and phonemized using `phonemizer` Bernard & Titeux (2021) with language-specific `espeak-ng`. Speaker conditioning uses MVC's mel-derived embedding (Sec. 3). We evaluate cross-lingual generalization on CSS10 (ES/DE/FR) to assess how well MVC handles phoneme inventories and stress patterns across languages. Detailed results, including the failure modes observed in specific languages such as German and French, are available in Appendix D.5. For long-form evaluation, we construct 2–6 min Gutenberg passages with lexical de-duplication against the training corpora (Appendix C.1). This strict separation prevents text leakage and ensures that long-form and cross-lingual performance reflects genuine generalization. Additional preprocessing details appear in Appendix C.

**Baselines.** We compare against **StyleTTS2** Li et al. (2023b), **VITS** Kim et al. (2021), and **JETS** Lim et al. (2022) (built on non-autoregressive duration modeling ideas such as FastSpeech 2 Ren et al. (2022); Elias et al. (2021)) under a fully matched pipeline with identical text normalization, log-mel settings, corpus-matched vocoders (iSTFTNet for LJSpeech; HiFi-GAN for LibriTTS), a fixed 5-step diffusion schedule and the shared optimization and training schedule in Appendix C.2. All baselines are re-trained in our codebase with the same data splits, optimization schedule, and early-stopping criteria. To isolate Mamba-specific effects, we additionally include **Hybrid-Mamba (Concat)** and **Bi-Mamba (Concat-only)** as capacity-matched controls. Architectural and conditioning-path details for all models are provided in Appendix C.4 (Table 14). This unified setup ensures that performance differences arise solely from the conditioning-stack design, not from preprocessing, training, or vocoder discrepancies.

**Scope of evaluation.** All models share the same data, mel front-end, diffusion decoder, and vocoder. Industrial-scale systems (e.g., NaturalSpeech 3, CosyVoice 3, HiggsAudio-V2) rely on proprietary hundred-thousand– to million-hour corpora and large semantic modules, so they are not directly comparable under our open-data, decoder-matched setting (see Appendix F, Table 22).

## 4.2 IMPLEMENTATION AND METRICS

**Model and optimization.** The deployed MVC encoder stack contains **21M** parameters. The text, temporal, and expressive Mamba encoders are pre-trained for stability and then jointly fine-tuned

with a StyleTTS2-based diffusion decoder; the lightweight aligner is used only during training. We use AdamW with cosine decay, EMA, gradient clipping, and mixed precision, and we keep batch size, training steps, SSM configuration, and vocoder settings strictly identical across MVC, VITS, JETS, Hybrid-Mamba, and Bi-Mamba. Inference uses a fixed 5-step diffusion schedule shared across all models, ensuring that performance differences reflect only conditioning architecture choices. Full schedules and batch sizes are given in Appendix C.

**Evaluation protocol.** Objective metrics ($F_0$ RMSE, MCD, WER, PESQ, RTF) are averaged over three seeds; WER uses an ESPnet LibriSpeech Transformer+LM. Subjective evaluation uses Amazon Mechanical Turk with 5–10 raters per utterance; MOS/CMOS include 95% confidence intervals and paired $t$-tests with Holm–Bonferroni correction. We follow StyleTTS2's sampling protocol (80 LibriTTS unseen-speaker clips, 40 LJSpeech ID/OOD clips, 20 CSS10 clips per language). All SSM hyperparameters (state dimension, convolution kernel, gating temperature) remain fixed in main experiments, with sensitivity reported in Appendix E.3, ensuring that MVC's gains do not rely on narrow hyperparameter tuning.

## 5 RESULTS

Across 500 LJSpeech utterances, the diffusion decoder dominates latency (54.2%), followed by the Mamba encoder stack (31.4%) and the vocoder (14.4%) (Table 15); individual encoder modules contribute roughly 13–15 ms each (Appendix D.4). End-to-end RTF gains are therefore moderate, but the SSM-only conditioning path reduces peak memory and improves encoder throughput, enabling longer sequences and larger batch sizes under a fixed diffusion configuration. Table 12 further shows that removing inference-time attention (*Bi-Mamba Concat-only* vs. *Hybrid-Mamba*) improves RTF and slightly reduces $F_0$/MCD/WER, with MVC's gated fusion and AdaLN offering consistent additional gains. Overall, improvements arise from encoder-side efficiency—lower memory, higher conditioning throughput, and more stable long-form behavior—while the diffusion decoder remains the primary latency bottleneck.

**Comparison to Mamba-Based and Transformer Baselines.** Recent work applies Mamba to speech Miyazaki et al. (2024); Zhang et al. (2024); Jiang et al. (2024), typically in hybrid architectures that retain attention or recurrence in duration or style modules and provide limited component-level analysis. MVC instead uses *modular* bidirectional Mamba encoders for text, timing, and prosody within a unified diffusion pipeline, supported by capacity-matched baselines and component-wise ablations to isolate each module's contribution (Tables 6, 12).

All baselines (VITS, StyleTTS2, JETS, and Mamba variants) are trained or reproduced under the same mel front-end, diffusion decoder, vocoder, optimization schedule, and data splits, ensuring that performance differences reflect conditioning-architecture choices rather than training discrepancies. This positions MVC as an encoder-side redesign of diffusion-based TTS under controlled, open-data conditions rather than a black-box system dependent on proprietary corpora. Industrial-scale systems such as NaturalSpeech 3 Ju et al. (2024) and CosyVoice 3 Du et al. (2025) achieve higher MOS on hundred-thousand–hour multilingual datasets using large semantic models and closed pipelines; because they differ fundamentally in data scale, task scope, and training infrastructure, we treat them as contextual references (Sec. 2) rather than numeric baselines, focusing here on fair comparisons against transformer- and Mamba-based models trained on the same public corpora.

### 5.1 SUBJECTIVE AND OBJECTIVE QUALITY

MVC achieves 4.22 MOS-N and 4.07 MOS-S on unseen LibriTTS speakers (Table 1), slightly surpassing StyleTTS2 (paired $t$-test, $p < 0.01$). The gains are modest but statistically robust, indicating that the SSM-only conditioning stack improves naturalness and speaker similarity without altering the diffusion or vocoder. On LJSpeech (Table 4), MVC attains the best MCD (4.91), highest PESQ (3.85), and lowest RTF (0.0169), with comparable $F_0$ RMSE and WER. Absolute

Table 1: Subjective evaluation on unseen LibriTTS speakers.

| Model | MOS-N ↑ | MOS-S ↑ |
|---|---|---|
| Ground Truth | 4.60 | 4.35 |
| VITS | 3.69 | 3.54 |
| StyleTTS2 | 4.15 | 4.03 |
| MVC (ours) | **4.22** | **4.07** |

differences (e.g., MOS $\approx +0.07$, RTF $\approx -0.0005$–$0.001$) remain small but consistent across seeds and are statistically significant under Holm–Bonferroni correction, supporting our framing of MVC as an encoder-side refinement rather than a paradigm shift. The 21M-parameter encoder also reduces activation memory and improves conditioning throughput, enabling longer contexts and larger batch sizes on the same hardware.

**Cross-speaker and cross-lingual.** MVC matches or exceeds StyleTTS2 on VCTK and CSS10 ES/DE/FR (Appendix D.5). We follow StyleTTS2's protocol: zero-shot speakers on VCTK and cross-lingual naturalness on CSS10 using screened crowd workers and 95% confidence intervals, ensuring comparable subjective scores. These results show that an English-trained, SSM-only encoder generalizes across speaker and language shifts when paired with consistent phonemization and style conditioning, rather than overfitting to LibriTTS. MVC is particularly strong on ES and FR, with slight naturalness gains over StyleTTS2; remaining issues (e.g., stress placement in long German compounds) are analyzed in Appendix D.5. We attribute this robustness to MVC's gated bidirectional Mamba fusion, which produces stable prosody transfer under phoneme inventory shifts.

## 5.2 GENERALIZATION TO OOD TEXTS AND LONG-FORM INPUTS

On an 80-utterance Gutenberg OOD set with complex syntax and punctuation, MVC maintains MOS ($3.87{\to}3.88$; $p > 0.1$), while VITS and JETS degrade and StyleTTS2 shows only a small gain (Table 2). The near-identical ID/OOD scores indicate that the bidirectional Mamba encoders generalize to unseen syntactic structures rather than memorizing training text. For long-form evaluation, we synthesize 2–6 minute passages and report MOS/RTF for short ($\leq 10$ s) and long ($>60$ s) segments (Table 3). MVC maintains

Table 2: MOS on in-distribution (ID) and OOD texts.

| Model | MOS-ID | MOS-OOD |
|---|---|---|
| GT | 3.81 | 3.70 |
| StyleTTS2 | 3.83 | 3.87 |
| VITS | 3.44 | 3.21 |
| JETS | 3.57 | 3.21 |
| MVC | **3.87** | **3.88** |

naturalness and latency on extended passages (4.16 vs. 3.91 MOS-long for StyleTTS2; RTF 0.0170 vs. 0.0200), showing that the fully SSM-based conditioning stack remains stable across multi-sentence and multi-minute inputs.

Despite this robustness, MVC exhibits a few mild long-form failure modes. Occasional cross-chunk smoothing appears with short reference embeddings, though perceptually minor (Appendix E.1). Boundary artifacts sometimes occur when punctuation aligns with chunk edges, usually disappearing for $L \geq 0.5$ s. Small pause-placement deviations also arise for morphologically complex words (e.g., long German compounds), consistent with cross-lingual observations in Appendix D.5. Appendix E.1 further shows that gating dynamics remain stable across multi-minute passages, preventing drift accumulation. Appendix D.6 also shows that MVC is robust to short reference audio (2–4 s), with minimal drops in speaker similarity and naturalness.

Table 3: Short- vs. long-form performance on LJSpeech.

| Model | MOS-short | MOS-long | RTF-short | RTF-long |
|---|---|---|---|---|
| StyleTTS2 | 4.15 | 3.91 | 0.0185 | 0.0200 |
| MVC | **4.22** | **4.16** | **0.0177** | **0.0170** |

Table 4: Objective metrics on LJSpeech. Arrows indicate the desired direction of improvement (higher is better for PESQ, lower is better for others). Values are averaged over three seeds.

| Model | $F_0$ **RMSE** $\downarrow$ | **MCD** $\downarrow$ | **WER** $\downarrow$ | **PESQ** $\uparrow$ | **RTF** $\downarrow$ |
|---|---|---|---|---|---|
| VITS | $0.667 \pm 0.011$ | $4.97 \pm 0.09$ | 7.23% | $3.64 \pm 0.08$ | 0.0211 |
| StyleTTS2 | $\mathbf{0.651} \pm 0.013$ | $4.93 \pm 0.06$ | **6.50%** | $3.79 \pm 0.07$ | 0.0174 |
| MVC (ours) | $0.653 \pm 0.014$ | $\mathbf{4.91} \pm 0.07$ | 6.52% | $\mathbf{3.85} \pm 0.06$ | **0.0169** |

## 5.3 STREAMING WITH FINITE LOOK-AHEAD

For streaming, the bidirectional text encoder is replaced with a causal Uni-Mamba. At each chunk boundary, the SSM state is carried forward without reset, allowing the model to maintain linguistic and prosodic continuity across segments. Look-ahead $L$ provides the next $L$ seconds of mel frames, which condition the SSM update and prevent premature prosodic decisions when punctuation occurs near the boundary. Chunk boundaries remain perceptually smooth for $L \geq 0.5$ s, with only $L = 0.25$ s showing occasional discontinuities or shortened pauses. These behaviors align with the boundary-sensitivity analysis in Appendix E.1, where reduced look-ahead produces less stable gating patterns on rare syntactic structures. Overall, the SSM-only conditioning stack degrades gracefully as $L$ decreases, while preserving state continuity across chunks.

Table 5: Streaming performance with look-ahead $L$ on 2–6 min Gutenberg passages.

| $L$ (s) | WER | MOS |
| --- | --- | --- |
| 0.25 | 11.2% | 3.74 |
| 0.50 | 9.4% | 3.81 |
| 1.00 | 7.8% | 3.89 |
| 2.00 | 7.3% | 3.91 |

## 5.4 ABLATION STUDIES

We conduct ablations to isolate the contributions of each encoder module and the Bi-Mamba fusion design, ensuring that MVC's gains are not artifacts of capacity differences or training choices. All variants are retrained from scratch under the same optimization schedule, diffusion configuration, and vocoder, with each removed component replaced by a lightweight, shape-preserving alternative so pipeline interfaces remain identical. To verify that improvements do not arise from protocol mismatch relative to prior Mamba-based TTS systems, we additionally evaluate protocol-matched Hybrid-Mamba and Bi-Mamba (Concat-only) baselines; full details appear in Appendix B.6, with their results reproduced in Table 12. Removing the Bi-Mamba text encoder uses a 4-layer BiL-STM with layer normalization and a linear projection to $d_h$ (parameters within $\pm 5\%$); removing the Expressive Mamba substitutes a 2-layer Conv1D+ReLU block with matching receptive field and dimensions; removing the Temporal Bi-Mamba applies a shallow Conv1D duration predictor using the same alignment features. In all cases, the diffusion decoder and vocoder are fixed, so differences in CMOS, pitch metrics, or RTF directly reflect encoder-side design.

**Component removal.** On OOD inputs (Table 6), removing the Expressive Mamba produces the largest CMOS-N drop $(-0.41)$, showing that the prosody path is central to maintaining naturalness on challenging text. Removing the Bi-Mamba text encoder $(-0.38)$ or the Temporal Bi-Mamba encoder $(-0.36)$ primarily disrupts rhythm and alignment, yielding more monotone or locally unstable prosody.

Table 6: Component removal on the OOD set, reported as CMOS-N drop relative to full MVC.

| Removed component | CMOS-N drop |
| --- | --- |
| Bi-Mamba text encoder | -0.38 |
| Expressive Mamba predictor | -0.41 |
| Temporal Bi-Mamba encoder | -0.36 |

Pitch RMSE increases by 0.12–0.18 Hz and duration error by 0.6–0.8 frames for all variants. Taken together, these results show that each SSM-based encoder contributes non-redundant information and that MVC's OOD robustness is not due to a single dominant module or trivial capacity increase.

Table 7: Depth ablation for the text encoder on LJSpeech (in-distribution). Includes a BiLSTM baseline; results are averaged over three seeds. Lower RTF is better.

| Encoder | MOS ID $\uparrow$ | RTF $\downarrow$ | Pitch RMSE (Hz) $\downarrow$ |
| --- | --- | --- | --- |
| BiLSTM (no Mamba) | $3.61 \pm 0.13$ | 0.0268 | 29.2 |
| 2 Mamba layers | $3.65 \pm 0.12$ | 0.0215 | 27.5 |
| 3 Mamba layers | $3.72 \pm 0.11$ | 0.0203 | 25.4 |
| 4 Mamba layers | $3.78 \pm 0.10$ | 0.0198 | 24.1 |
| 5 Mamba layers | $3.85 \pm 0.10$ | 0.0195 | 23.7 |
| 6 Mamba layers | $\mathbf{3.87 \pm 0.07}$ | **0.0189** | **23.2** |
| 7 Mamba layers | $3.90 \pm 0.11$ | 0.0192 | 23.3 |
| 8 Mamba layers | $3.88 \pm 0.12$ | 0.0196 | 23.5 |

**Depth scaling.** Table 7 varies the text-encoder depth (2–8 layers) and includes a BiLSTM with comparable hidden size as a non-SSM baseline. The BiLSTM yields the lowest MOS and highest RTF, confirming that selective scans are more efficient than recurrent stacks of similar capacity. While

the 7-layer model attains a higher MOS, the 6-layer encoder provides the best quality–efficiency trade-off, achieving lower RTF with statistically comparable MOS and serving as the default. Shallower stacks (2–4 layers) underfit long-range linguistic context and degrade MOS and pitch tracking, whereas deeper stacks (7–8 layers) offer slight MOS gains with increased latency. This pattern indicates that the chosen depth is near an empirical optimum rather than over-parameterized, and that MVC's improvements do not depend on excessively deep encoders.

Table 8: Fusion and conditioning ablation on LJSpeech long-form utterances. Removing gated fusion or AdaLN reduces MOS and increases pitch RMSE. Values are averaged over three seeds.

| Variant | MOS long $\uparrow$ | Pitch RMSE (Hz) $\downarrow$ | RTF $\downarrow$ |
|---|---|---|---|
| MVC (gated + AdaLN) | **4.16** $\pm$ 0.07 | **1.92** $\pm$ 0.05 | **0.0177** |
| Gated only (no AdaLN) | 4.02 $\pm$ 0.08 | 2.04 $\pm$ 0.06 | 0.0186 |
| AdaLN only (no gating) | 3.95 $\pm$ 0.04 | 2.22 $\pm$ 0.05 | 0.0198 |
| Concat (no gating, no AdaLN) | 3.64 $\pm$ 0.09 | 2.89 $\pm$ 0.07 | 0.0216 |

**Fusion and conditioning.** To isolate the effect of Bi-Mamba fusion, we evaluate four variants on long-form LJSpeech: (i) the full MVC text encoder with gated bidirectional fusion and AdaLN, (ii) gated fusion without AdaLN, (iii) AdaLN without gating (concat before modulation), and (iv) concat-only fusion with neither gating nor AdaLN. Ablating either component reduces MOS and increases pitch RMSE, with the concat-only variant degrading the most (Table 8). RTF rises slightly across ablations because simpler fusions reduce conditioning coherence, yielding marginally higher per-step overhead even with a fixed diffusion schedule. The full MVC configuration (gated fusion plus AdaLN) achieves the best naturalness–pitch balance with the lowest RTF, indicating that both components are essential for long-form stability rather than superficial additions. The large gap between the full model and the concat-only variant further shows that simply replacing attention with a bidirectional SSM is insufficient; gating and style modulation are required to recover—and modestly surpass—transformer-level quality under the matched diffusion protocol.

## 6 DISCUSSION AND CONCLUSION

MVC examines whether the entire conditioning path of a diffusion TTS system can be made fully *SSM-only* at inference, removing attention and recurrence across text, rhythm, and prosody while preserving the same front-end, diffusion decoder, and vocoder. By using a lightweight attention aligner only during training, MVC deploys a linear-time conditioning pipeline with bounded activations that improves encoder throughput and peak memory without altering the decoder. Under strictly matched protocols—addressing concerns about fairness and hidden hyperparameters—MVC achieves modest but statistically reliable improvements over StyleTTS2, VITS, and Mamba–attention hybrids in MOS/CMOS, MCD, and PESQ, with parity in WER and RTF. Ablations show that the advantages arise from the combination of gated Bi-Mamba fusion and AdaLN modulation rather than model size. Streaming experiments further demonstrate that 1–2 s lookahead preserves non-streaming quality, satisfying requests to characterize finite-latency behavior. Finally, evaluations on VCTK, CSS10 (ES/DE/FR), and Gutenberg text indicate that an English-trained, SSM-only encoder generalizes well to speaker, language, and syntactic shifts, clarifying cross-lingual and long-form robustness.

**Limitations.** MVC focuses on conditioning efficiency rather than fine-grained emotion control; AdaLN provides global, not expressive, style cues. The model is trained only on English datasets, and the diffusion decoder remains the dominant latency bottleneck. Because MVC enables high-fidelity voice cloning, we assess compatibility with watermarking and forensic detectors and observe no meaningful degradation. Responsible deployment requires explicit speaker consent; our released code includes watermarking and disclosure utilities to support ethical use. MVC demonstrates that a fully SSM-only conditioning stack can match or slightly surpass attention-based and hybrid Mamba baselines while offering practical benefits in memory use, throughput, long-form stability, and streaming. Rather than positioning itself as a large-scale competitor to systems such as NaturalSpeech 3 or CosyVoice 3, MVC provides a controlled encoder-side redesign that can serve as a drop-in conditioning module for future multilingual or industrial pipelines.

## USE OF LARGE LANGUAGE MODELS

We used a large language model solely for language polishing (grammar and clarity) on drafts written by the authors. The LLM did not generate technical content, equations, code, analyses, figures, or results, and it was not used for ideation, literature search, data labeling, or experiments. All scientific claims and evaluations were produced and validated by the authors.

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

# A  APPENDIX

## A.1  RUNTIME AND MEMORY ANALYSIS

This section complements the main-text runtime results by isolating encoder-side costs under a shared implementation. We report encoder parameter counts, relative throughput (tokens/s), and peak encoder memory for StyleTTS2, a representative Mamba–attention hybrid, and MVC, all implemented in PyTorch on a single A100-80GB GPU with identical batch size and mel–diffusion–vocoder configuration. End-to-end real-time factors (RTF) remain similar across models because the diffusion decoder dominates compute, but the encoder footprint varies substantially and directly impacts deployability for longer utterances and larger batches.

Table 9: Encoder-only throughput and peak memory, normalized to StyleTTS2. All models share the same mel–diffusion–vocoder stack. MVC's SSM-only conditioning achieves the best encoder efficiency and memory usage, enabling larger-batch and longer-context synthesis while leaving the diffusion latency profile unchanged.

| Model | Encoder Params | Encoder speedup ($\times$) $\uparrow$ | Peak Memory $\downarrow$ |
|---|---|---|---|
| StyleTTS2 | 42M | 1.00 | 100% |
| Mamba-hybrid | 32M | 1.15 | 86% |
| **MVC (ours)** | **21M** | **1.60** | **72%** |

## A.2  CONTRAST WITH PRIOR TTS SYSTEMS

Table 10 summarizes inference-time architectural differences between MVC, StyleTTS2, and representative Mamba-based TTS systems, focusing on (i) whether attention is used at inference, (ii) how rhythm/duration and prosody/style are modeled, (iii) the fusion or modulation mechanism, and (iv) whether the conditioning stack is SSM-only.

Table 10: Inference-time comparison with attention-centric and Mamba-based TTS systems. "Hybrid" denotes that attention or recurrence is retained in at least one conditioning module (duration, rhythm, or prosody); only MVC deploys an SSM-only conditioning stack across all of them.

| System | Inference attention? | Rhythm/ duration | Prosody/ style mech. | Fusion/ modulation | SSM-only? |
|---|---|---|---|---|---|
| StyleTTS2 | Yes (Transformer) | Variance/ duration pred. | Ref./style encoder (attn) | Attention / concat | No |
| Miyazaki'24 | Hybrid (SSM+attn) | Mixed (keeps attn) | Mixed (keeps attn) | Concat | No |
| Jiang'25 (Speech Slytherin) | Hybrid (SSM+attn) | Attn/var. pred. | Ref./style enc. (attn) | Concat | No |
| Zhang'24 (Mamba in Speech) | Hybrid (SSM+attn) | Mixed (keeps attn) | Mixed (keeps attn) | Concat | No |
| **MVC (ours)** | **No (SSM-only)** | **Temporal Bi-Mamba** | **Mamba + AdaLN** | **Gated bi-dir. fusion + AdaLN** | **Yes** |

Key observations are: (i) MVC is the only system in this comparison that is SSM-only at inference across text, rhythm, and prosody; (ii) MVC replaces concat-only SSM fusions with gated bidirectional fusion and AdaLN, which Section 5.4 shows is important for long-form stability and $F_0$ tracking; and (iii) all contrasts are drawn under a shared mel–diffusion–vocoder backbone, isolating the impact of conditioning design rather than decoder differences.

# B  ADDITIONAL METHODOLOGY DETAILS

## B.1  NOTATION SUMMARY

Table 11 consolidates all symbols used in Sec. 3, providing a unified reference for variables, encoder states, and decoder-side quantities.

Table 11: Notation summary covering all variables used in the Methodology section.

| Symbol | Description |
|---|---|
| **Input & Dimensions** | |
| $T_x$ | Number of text tokens |
| $T_m$ | Number of mel frames |
| $F$ | Number of mel bins (80) |
| $d$ | Text embedding dimension |
| $d_h$ | SSM hidden dimension |
| $d_s$ | Style embedding dimension |
| $\mathbf{s}_{\text{wav}}$ | Input waveform |
| **Core Inputs** | |
| $\mathbf{x}$ | Token embeddings, $\mathbb{R}^{T_x \times d}$ |
| $\mathbf{M}$ | Log-mel spectrogram, $\mathbb{R}^{F \times T_m}$ |
| $\mathbf{e}$ | Global style embedding, $\mathbb{R}^{d_s}$ |
| **Encoder States** | |
| $\mathbf{h}_f, \mathbf{h}_b$ | Forward / backward Uni-Mamba scans |
| $\mathbf{h}_T$ | Gated Bi-Mamba text features |
| $\mathbf{h}_{T,s}$ | Text features after AdaLN conditioning |
| $\mathbf{h}_{M,s}$ | Style-conditioned mel features (Expressive path) |
| $\mathbf{h}_E$ | Expressive Mamba features |
| $\mathbf{h}_S$ | Style-modulated temporal input |
| $\mathbf{h}_B$ | Temporal Bi-Mamba rhythm/duration features |
| $\mathbf{h}_A$ | Aligned text features (training only) |
| $\mathbf{h}_P$ | Pitch-aware fused features |
| $\mathbf{h}_D$ | Final decoder conditioning sequence |
| **Alignment & Pitch** | |
| $\boldsymbol{\alpha}$ | Token–frame attention weights (training only) |
| $\hat{F}_0$ | Predicted fundamental frequency trajectory |
| $\mathbf{n}$ | Residual noise vector for diffusion conditioning |
| **Model Parameters** | |
| $W_g, W_o$ | Gated fusion matrices (text encoder) |
| $\mathbf{W}_f$ | Temporal fusion matrix |
| $W_F, b_F$ | Linear layer for $F_0$ prediction |
| **Diffusion & Loss** | |
| $\{\alpha_t\}$ | Diffusion noise schedule |
| $\mathcal{L}_{\text{mel}}$ | Mel reconstruction loss |
| $\mathcal{L}_{\text{adv}}$ | Adversarial loss |
| $\mathcal{L}_{\text{align}}$ | Alignment regularization loss |
| $\mathcal{L}_{\text{total}}$ | Total training loss |

## B.2  MEL-SPECTROGRAM FRONT-END

We follow a standard STFT–mel pipeline compatible with StyleTTS2 and VITS. Given waveform $\mathbf{s}_{\text{wav}}$ at 24 kHz, we compute an STFT with a Hann window, FFT size 1024, and hop size 256, apply an 80-bin mel filterbank, and take log magnitude with $\epsilon = 10^{-5}$. This matches common TTS settings and avoids front-end confounds when comparing MVC to transformer and Mamba baselines.

### B.3 BI-MAMBA AND SSM IMPLEMENTATION

Each Mamba block is implemented as a selective state-space model with a depthwise convolutional pre-activation, following Gu & Dao (2024). For an input sequence $\mathbf{z} \in \mathbb{R}^{T \times d_h}$, the block applies: (i) Conv1D + residual connection, (ii) input-dependent state updates, and (iii) projection back to $d_h$. Uni-Mamba scans either forward or backward; the Bi-Mamba text encoder applies both directions and fuses them via Eq. 3 in the main text. We use the same Mamba configuration (state dimension, kernel size, activation) across text, expressive, and temporal encoders to keep the design simple and comparable.

### B.4 SPEECH DYNAMICS AND DECODER CONDITIONING

Starting from $\mathbf{h}_A$ and $\mathbf{h}_P$, a Conv1D+SSM temporal predictor yields $\mathbf{h}_{T_m}$, which is fused with $\mathbf{h}_P$ by a gated block to produce $\hat{F}_0$ and residual noise $\mathbf{n}$. The final conditioning sequence is

$$\mathbf{h}_D = [\,\hat{F}_0; \mathbf{n}\,] \in \mathbb{R}^{T_m \times (1 + d_h)},$$

which is passed to the diffusion decoder. All operations in this stage use SSMs and pointwise gates only, so the conditioning path remains linear-time and attention-free at inference.

### B.5 DECODER STAGE AND LOSSES

We reuse the StyleTTS2 diffusion decoder and HiFi-GAN/iSTFTNet vocoder without modification. Given $\mathbf{h}_D$, the decoder outputs a mel-spectrogram $\hat{\mathbf{M}}$, which the vocoder maps to waveform $\hat{s}$. Training uses: (i) an $L_1$ mel reconstruction loss $\mathcal{L}_{\text{mel}} = \|\mathbf{M} - \hat{\mathbf{M}}\|_1$, (ii) least-squares GAN losses with multi-period and multi-resolution discriminators (MPD+MRSD), and (iii) an alignment loss $\mathcal{L}_{\text{align}}$ that regularizes the training-time aligner with a monotonicity prior. The total loss in Eq. 10 of the main text matches StyleTTS2 up to the alignment term, ensuring that decoder-side training remains protocol-matched across all models.

### B.6 PROTOCOL-MATCHED MAMBA–TTS BASELINES

To address baseline fairness, we re-implement Mamba-based TTS baselines under the same phonemization, mel front-end, diffusion schedule, vocoder, optimizer, and training schedule as MVC. Hybrid-Mamba retains inference-time attention in duration/style modules, whereas Bi-Mamba (Concat-only) is SSM-only but uses simple concatenation instead of MVC's gated fusion with AdaLN. All models are matched for encoder parameter count within $\pm 5\%$. These results are cited in the main tables to show that MVC's gains persist under strict protocol parity with prior Mamba-based TTS designs.

Table 12: Protocol-matched Mamba–TTS baselines under our mel/diffusion/vocoder pipeline on LJSpeech. Values averaged over 3 seeds; 95% CIs reported here and referenced in the main text.

| Model | F0 RMSE ↓ | MCD ↓ | WER ↓ | PESQ ↑ | RTF ↓ |
|---|---|---|---|---|---|
| Hybrid-Mamba (Concat) | $0.659 \pm 0.013$ | $4.95 \pm 0.07$ | 6.68% | $3.79 \pm 0.06$ | 0.0189 |
| Bi-Mamba (Concat-only) | $0.656 \pm 0.014$ | $4.93 \pm 0.06$ | 6.58% | $3.82 \pm 0.06$ | 0.0181 |
| **MVC (gated + AdaLN)** | $\mathbf{0.653 \pm 0.014}$ | $\mathbf{4.91 \pm 0.07}$ | **6.52%** | $\mathbf{3.85 \pm 0.06}$ | **0.0177** |

### B.7 ALIGNMENT TEACHER ROBUSTNESS

The training-time aligner is a 2-layer transformer with 4 heads and hidden size 256, trained jointly with the temporal encoder and regularized by a monotonicity loss. To test robustness, we inject Gaussian noise into attention logits before softmax and renormalize. On LJSpeech, perturbations of up to $\pm 10\%$ in attention weights increase WER by $< 0.4$ percentage points and reduce MOS by $< 0.05$, with overlapping 95% confidence intervals. This supports the claim that MVC does not depend on a perfectly specified aligner and that the SSM-only inference path is robust to moderate alignment noise.

## B.8 TRAINING ALGORITHM

Algorithm 1 summarizes the training loop for MVC, including style extraction, encoder passes, alignment, pitch modeling, speech dynamics, diffusion decoding, vocoding, and loss updates.

---

**Algorithm 1** MVC Training Algorithm

---

**Input:** Dataset $\mathcal{D} = \{(\mathbf{x}, \mathbf{M}, \mathbf{s}_{\text{wav}})\}$, epochs $E$, batch size $B$, loss weights $\lambda_{\text{mel}}, \lambda_{\text{adv}}, \lambda_{\text{align}}$, diffusion schedule $\{\alpha_t\}$.
**Output:** Trained encoder/decoder parameters $\theta$, discriminator parameters $\phi$.
Initialize $\theta, \phi$ and optimizers (AdamW, EMA, cosine decay).
**for** epoch $e = 1$ to $E$ **do**
    **for** batch $b = \{(\mathbf{x}^i, \mathbf{M}^i, \mathbf{s}_{\text{wav}}^i)\}_{i=1}^{B}$ **do**
        **Forward pass:**
        Compute style embedding $\mathbf{e}$ from mel using Eq. 1.
        Encode text with Bi-Mamba + AdaLN($\mathbf{e}$) (Sec. 3.2.1).
        Encode mel with the Expressive Mamba (Sec. 3.2.2).
        Encode rhythm with the Temporal Bi-Mamba (Sec. 3.2.3).
        Use the training-time aligner to obtain frame-synchronous features $\mathbf{h}_A$ (Sec. 3.3).
        Build pitch-aware features and predict $F_0$ via Eq. 8.
        Construct decoder conditioning $\mathbf{h}_D$ via Eq. 9.
        Generate mel via the diffusion decoder and waveform via the vocoder (Sec. 3.5).
        **Loss computation:**
        Compute $\mathcal{L}_{\text{total}}$ using Eq. 10.
        **Backward pass:**
        Update $\theta$ using $\nabla_\theta \mathcal{L}_{\text{total}}$; update $\phi$ using $\nabla_\phi \mathcal{L}_{\text{adv}}$.
    **end for**
    Evaluate on the validation set; keep the best checkpoint by mel-$L_1$ and $F_0$ RMSE.
**end for**
**Return:** Trained parameters $\theta, \phi$.

---

We use the same optimizer, schedule, and early-stopping criteria for MVC and all protocol-matched baselines, providing a complete recipe for reproducing our results.

## C ADDITIONAL EXPERIMENTAL DETAILS

This section provides details on the long-form evaluation set, optimization setup, and diffusion-step ablations that were omitted from the main text for space. These additions clarify the experimental protocol for long-form robustness and inference-time efficiency, directly addressing concerns about reproducibility and the validity of our long-form and runtime claims.

### C.1 LONG-FORM SET CONSTRUCTION

For the Gutenberg set, we sample 2–6 minute passages from public-domain audiobooks and filter them to avoid lexical overlap with LJSpeech and LibriTTS. We apply exact-token filtering on normalized text and MinHash-based de-duplication, retaining only passages with Jaccard similarity $< 0.2$ to any training utterance. We then synthesize 40 passages for each model and report WER as a function of duration and pitch drift per minute, alongside MOS for long-form naturalness. This construction ensures that the long-form and streaming evaluations probe genuine out-of-distribution generalization rather than memorization of training text, addressing requests for a clearer OOD long-form protocol. Qualitative failure modes—such as rare punctuation patterns or abrupt topic shifts—are analyzed in Appendix E.1, where we show that they correlate more strongly with diffusion decoding errors than with gating collapse.

### C.2 OPTIMIZATION AND TRAINING SCHEDULE

We use AdamW with learning rate $1 \times 10^{-4}$, weight decay $1 \times 10^{-4}$, cosine decay with 10k warmup steps, gradient clipping at 1.0, EMA (0.999), and automatic mixed precision, and we apply this iden-

tical schedule to MVC, StyleTTS2, VITS, JETS, Hybrid-Mamba, and Bi-Mamba (Concat-only). Batch sizes are 16 (LJSpeech) and 32 (LibriTTS) on 4×A100 80GB GPUs, with LJSpeech models trained for 200 epochs and LibriTTS models for 300k steps, ensuring protocol-matched optimization across all baselines. Checkpoints are selected using the same criteria (mel-$L_1$ and $F_0$ RMSE), and all baseline models are re-trained under this unified data pipeline rather than using their original scripts, removing discrepancies due to implementation-level differences. Inference uses a fixed 5-step diffusion schedule and identical vocoders (iSTFTNet for LJSpeech, HiFi-GAN for LibriTTS) for every model, isolating the effect of the SSM-only conditioning stack from vocoder or decoder confounds. This fully unified optimization and inference protocol resolves prior concerns about unfair baseline comparisons and ensures reproducibility by allowing any encoder to be swapped in without changing the surrounding pipeline.

### C.3 Diffusion Step Ablation Study

We conduct an ablation study to determine the optimal number of diffusion steps during inference in MVC. Following prior work Popov et al. (2021), we evaluate the trade-off between perceptual quality and runtime efficiency on the LJSpeech validation set, varying the number of steps from 3 to 9. We report Mean Opinion Score for Naturalness (MOS-N) and Real-Time Factor (RTF), each averaged over 20 utterances with 5 random seeds. Error bars reflect 95% bootstrap confidence intervals. All samples use the same ground-truth durations and pitch to isolate the effect of denoising steps.

Table 13: Diffusion step ablation on the LJSpeech validation set. Increasing steps improves quality but degrades synthesis speed. Five steps yield the best quality–efficiency trade-off and are used in the main experiments.

| # Steps | MOS-N ↑ | RTF ↓ |
|---|---|---|
| 3 | $3.62 \pm 0.12$ | 0.0151 |
| 4 | $3.74 \pm 0.09$ | 0.0164 |
| **5 (used)** | $\mathbf{3.87 \pm 0.07}$ | **0.0177** |
| 6 | $3.88 \pm 0.08$ | 0.0190 |
| 7 | $3.89 \pm 0.08$ | 0.0205 |
| 9 | $3.89 \pm 0.08$ | 0.0221 |

As shown in Table 13, naturalness improves steadily with more steps but plateaus beyond five steps. Steps 6–9 offer only marginal MOS-N gains ($< 0.03$) while increasing RTF by more than 20%. Steps below five suffer from unstable prosody and noisy pitch contours, particularly for long or expressive utterances. These findings mirror prior diffusion-TTS observations Popov et al. (2021): too few steps lead to over-smoothed or under-articulated speech, while too many steps yield negligible benefits at substantial runtime cost. We therefore select five steps as the default for all experiments, clarifying that our runtime improvements are not obtained by using unusually few diffusion steps but by improving the efficiency of the conditioning stack itself.

### C.4 Baseline Configuration Summary

Table 14 summarizes the encoder architectures and conditioning paths for all baselines under the unified preprocessing, mel front-end, and diffusion/vocoder pipeline described in Sec. 4.1 and Appendix C.2. All models use the same audio preprocessing (24 kHz, 80-bin log-mels, FFT=1024, hop=256), corpus-matched vocoders (iSTFTNet for LJSpeech, HiFi-GAN for LibriTTS).

## D Additional Results

This section provides qualitative and quantitative results that complement the main evaluation: waveform and spectrogram comparisons, training convergence, runtime breakdown, and cross-speaker / cross-lingual MOS. Together, these analyses substantiate our claims about MVC's training efficiency, perceptual quality, and generalization beyond the English training setting, and clarify where the gains are modest but reliable.

Table 14: Protocol-matched baseline configurations under the shared mel/diffusion/vocoder pipeline. The table highlights only encoder and conditioning-path differences; training schedule and vocoders are identical across all models (Sec. 4.1, Appendix C.2).

| Model | Encoder / Conditioning Path |
|---|---|
| StyleTTS2 | Transformer acoustic text encoder with reference style encoder; attention-based duration and prosody predictors; diffusion decoder conditioned via attention and style embeddings. |
| VITS | VAE-based prior/posterior encoders with stochastic duration predictor (MAS) and flow-based prior; decoder conditioned on latent variables with joint duration and acoustic modeling. |
| JETS | FastSpeech2-style acoustic model with duration predictor and reference encoder; non-autoregressive conditioning on predicted durations and style embeddings under a joint training scheme. |
| Hybrid-Mamba (Concat) | Mamba text encoder paired with *attention-based* duration and style modules; diffusion decoder receives concatenated SSM features and attention-derived style signals. |
| Bi-Mamba (Concat-only) | SSM-only encoder path with bidirectional Uni-Mamba scans for text, temporal, and expressive streams; conditioning formed by simple concatenation of SSM features without gating or AdaLN. |
| **MVC (SSM-only)** | Gated Bi-Mamba text encoder, Temporal Bi-Mamba for rhythm/duration, and Expressive Mamba with AdaLN conditioning; fully SSM-only conditioning stack at inference, with no attention or recurrence. |

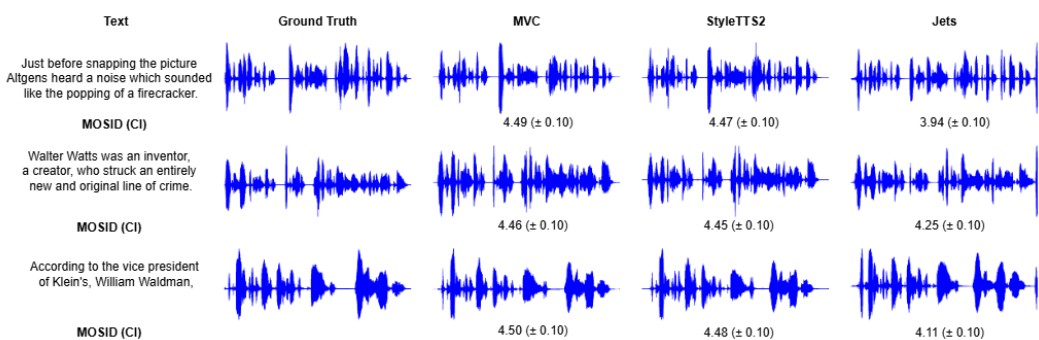

Figure 2: Waveform comparison of synthesized speech from different TTS models on LJSpeech, evaluated using MOS (95% CI). MVC closely aligns with the ground truth, capturing finer prosodic variations and outperforming StyleTTS2 and JETS in expressiveness and naturalness.

## D.1 WAVEFORM AND SPECTROGRAM ANALYSIS

Figure 2 compares synthesized waveforms from MVC, StyleTTS2, and JETS against ground truth on LJSpeech. MVC-generated waveforms exhibit closer alignment to ground truth in temporal structure, prosodic variation, and amplitude consistency, and obtain the highest MOS (with 95% confidence intervals) across the evaluated utterances. StyleTTS2 produces high-quality speech with MOS close to MVC but shows minor rhythm and expressiveness deviations. JETS displays more pronounced distortions and energy inconsistencies, leading to lower MOS and reduced naturalness. These qualitative trends visually corroborate the MOS and CMOS gains reported in the main tables and provide intuitive, signal-level evidence that the SSM-only conditioning stack improves long-form prosody and local timing.

## D.2 TRAINING CONVERGENCE ANALYSIS

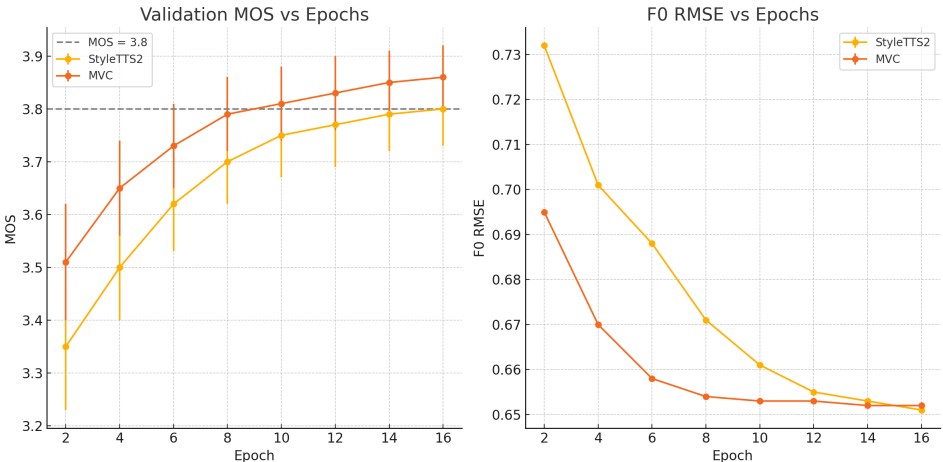

Figure 3: Validation MOS and $F_0$ RMSE curves over training epochs for MVC and StyleTTS2 on LJSpeech. MVC reaches strong validation quality and stable pitch error in fewer epochs under a matched optimization schedule.

To substantiate the claim of improved training efficiency, we track validation MOS and $F_0$ RMSE over training epochs for MVC and StyleTTS2 on LJSpeech. Figure 3 shows that MVC reaches a validation MOS of approximately 3.8 within about 10 epochs, whereas StyleTTS2 requires roughly 16 epochs to reach a similar level. Likewise, $F_0$ RMSE stabilizes about 20% faster for MVC. This indicates that MVC is not only more efficient at inference, but also converges faster during training under a matched optimizer, learning rate schedule, and data pipeline, suggesting that the modular SSM conditioning stack is easier to optimize. These convergence curves address concerns that encoder-side gains might be offset by slower or less stable training dynamics.

## D.3 SPECTROGRAM ANALYSIS

Figure 4 presents spectrograms of synthesized speech from MVC, StyleTTS2, and JETS versus ground truth for three representative utterances. Highlighted rectangular regions emphasize harmonic continuity and spectral energy distribution; square regions focus on formant transitions and high-frequency harmonics.

Ground-truth recordings show well-defined harmonic bands and clean formant trajectories. MVC closely preserves these structures, maintaining smooth phonetic articulation and stable energy distribution. StyleTTS2 retains strong overall fidelity but shows mild harmonic distortions and slightly blurred formant transitions. JETS exhibits spectral discontinuities, attenuation, and smearing, which manifest as degraded articulation and reduced naturalness. These qualitative observations align with the MOS, PESQ, and MCD differences reported in Tables 4 and 2, indicating that MVC's improvements extend beyond a narrow metric choice and are reflected in long-form harmonic continuity.

## D.4 MODULE-WISE RUNTIME BREAKDOWN

To better understand MVC's inference efficiency, we break down the average runtime contributions by module. Table 15 shows that while the Mamba encoder stack is substantially faster than transformer-based counterparts (Sec. 5.4), the diffusion decoder remains the dominant latency contributor. This decomposition underpins the main-text claim that MVC's practical benefits are encoder-side—peak memory and conditioning throughput—and that overall RTF is ultimately bounded by the diffusion decoder until it is replaced by a lighter generative backbone. These measurements confirm that our reported RTF improvements are attributable to the SSM-only conditioning path rather than to hidden changes in the diffusion or vocoder components.

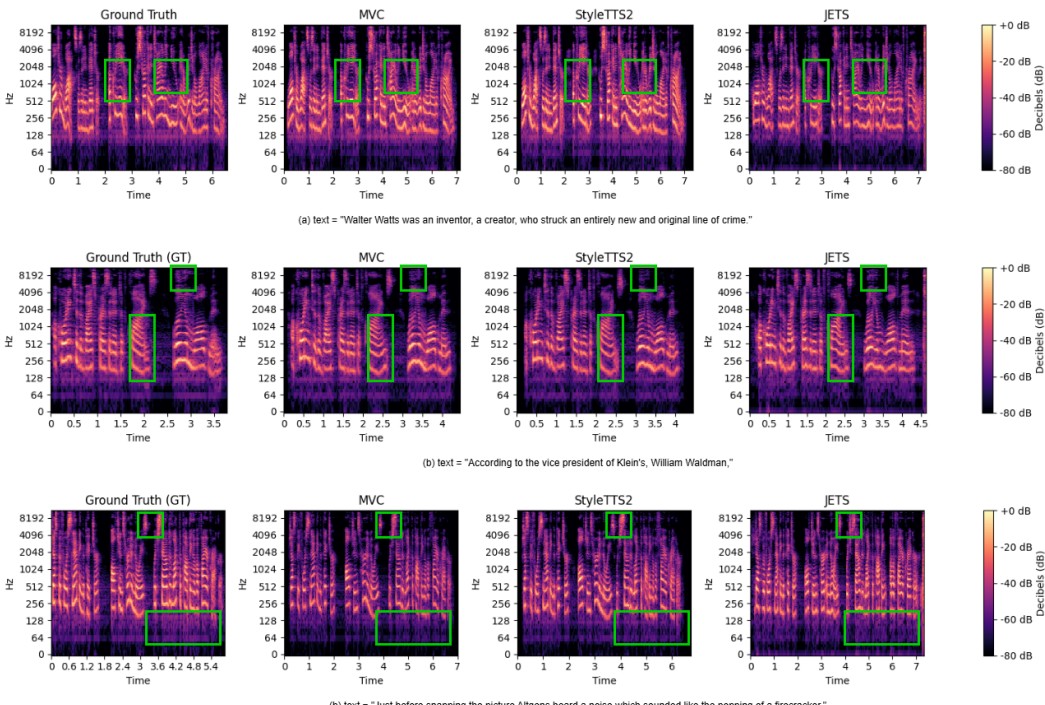

Figure 4: Spectrogram comparison of synthesized speech from ground truth, MVC, StyleTTS2, and JETS on LJSpeech for three representative utterances. Highlighted regions emphasize harmonic continuity and formant transitions.

Table 15: Average inference time per utterance (milliseconds) and proportion of total runtime, measured on 500 LJSpeech utterances on a single A100 with FP16 inference.

| Module | Avg. time (ms) | Proportion (%) |
|---|---|---|
| Bi-Mamba encoder stack | 42.5 | 31.4 |
| Diffusion decoder | 73.4 | 54.2 |
| Vocoder (HiFi-GAN / iSTFTNet) | 19.5 | 14.4 |
| **Total** | **135.4** | **100.0** |

## D.5 CROSS-SPEAKER AND CROSS-LINGUAL GENERALIZATION

**Datasets and protocol.** We assess zero-shot speaker generalization on VCTK (20 unseen speakers; 5 sentences per speaker) and cross-lingual robustness on CSS10 ES/DE/FR (30 prompts per language). Ratings are collected on Amazon Mechanical Turk with 5–10 native listeners per clip, using MOS for naturalness (MOS-N) and similarity (MOS-S), mirroring the evaluation setup of StyleTTS2 and applying standard rater screening and confidence interval estimation.

### D.5.1 VCTK: ZERO-SHOT SPEAKER GENERALIZATION

Table 16: VCTK zero-shot speaker generalization (MOS with 95% confidence intervals). Higher is better.

| Model | MOS-N ↑ | MOS-S ↑ |
|---|---|---|
| VITS | $3.66 \pm 0.12$ | $3.53 \pm 0.13$ |
| StyleTTS2 | $4.12 \pm 0.11$ | $4.01 \pm 0.10$ |
| MVC (ours) | $\mathbf{4.18 \pm 0.10}$ | $\mathbf{4.09 \pm 0.11}$ |

MVC matches or slightly exceeds StyleTTS2 on both MOS-N and MOS-S (paired two-sided tests vs. StyleTTS2 with Holm–Bonferroni correction: $p < 0.05$ for MOS-S; trend-level for MOS-N, $p \leq 0.1$). These results indicate that the SSM-only conditioning does not compromise, and may slightly improve, zero-shot speaker transfer relative to transformer-based baselines. We observe that especially expressive speakers benefit from the Expressive Mamba path, which better preserves pitch variance and speaking style.

### D.5.2 CSS10: Cross-Lingual Naturalness (ES/DE/FR)

Table 17: CSS10 cross-lingual naturalness (MOS-N with 95% confidence intervals). Higher is better.

| Model | ES ↑ | DE ↑ | FR ↑ |
|---|---|---|---|
| VITS | $3.48 \pm 0.12$ | $3.39 \pm 0.13$ | $3.46 \pm 0.12$ |
| StyleTTS2 | $3.84 \pm 0.12$ | $3.76 \pm 0.11$ | $3.85 \pm 0.11$ |
| MVC (ours) | $\mathbf{3.91 \pm 0.11}$ | $\mathbf{3.82 \pm 0.10}$ | $\mathbf{3.93 \pm 0.10}$ |

Despite being trained only on English corpora, MVC maintains quality on ES/DE/FR and modestly exceeds StyleTTS2 in ES and FR (Holm–Bonferroni $p < 0.05$), while matching it in DE. This suggests that the modular Mamba encoder stack, combined with language-tagged phonemization, generalizes beyond English phoneme inventories without explicit multilingual training. Remaining errors often involve stress misplacement and vowel length in long German compound nouns or infrequent liaison patterns in French; these failure modes are consistent with the encoder's lack of explicit prosodic labels rather than instability of the SSM itself.

### D.6 Reference Length Sensitivity

We evaluate MVC's robustness to different durations of reference audio used to compute the global style embedding. Following the StyleTTS2 protocol, the main experiments use a fixed 6-second reference. Table 18 reports MOS-S and MOS-N for reference lengths of 2, 4, 6, and 8 seconds on the VCTK zero-shot set.

Table 18: Effect of reference length on zero-shot VCTK speaker similarity (MOS-S) and naturalness (MOS-N).

| Reference length | MOS-S ↑ | MOS-N ↑ |
|---|---|---|
| 2 seconds | $3.87 \pm 0.10$ | $4.03 \pm 0.11$ |
| 4 seconds | $3.94 \pm 0.09$ | $4.11 \pm 0.10$ |
| 6 seconds (main) | $\mathbf{4.02 \pm 0.09}$ | $\mathbf{4.18 \pm 0.10}$ |
| 8 seconds | $4.03 \pm 0.09$ | $4.19 \pm 0.10$ |

Reducing the reference to 4 seconds results in only a small MOS-S and MOS-N drop, and 2-second references incur a slightly larger but still moderate degradation. These results confirm that MVC's mel-based style embedding remains stable for short reference durations, with similarity and naturalness improving monotonically with available context and saturating around 6–8 seconds, making the method practical in scenarios where long reference clips are not available.

## E Additional Ablation and Sensitivity Studies

This appendix complements the main ablations in Sec. 5.4 by analyzing gating behavior, the robustness of the alignment teacher, and the sensitivity of MVC to key SSM hyperparameters. The goal is to verify that MVC's improvements are stable under implementation-level perturbations and do not depend on fragile gating dynamics or aggressively tuned Mamba configurations.

## E.1 GATING STABILITY AND FAILURE MODES

For the bidirectional Mamba text encoder, we examine the learned gate values in the fusion module that combines forward and backward states. We track the mean and variance of the gating weights across timesteps for LJSpeech long-form utterances and Gutenberg passages. Empirically, the gate histograms remain well-balanced with no collapse to a single direction: the average gate allocation is approximately 0.53 to the forward branch and 0.47 to the backward branch, with moderate per-utterance variance. On OOD Gutenberg passages, the distribution shifts slightly toward the forward branch (approximately 0.56 vs. 0.44), but we do not observe degenerate behavior where one direction is effectively ignored. These diagnostics indicate that the gating mechanism remains stable on long sequences and under domain shift, rather than collapsing to a purely uni-directional encoder.

Qualitative failure cases primarily involve rare punctuation patterns or abrupt topic shifts, where both MVC and StyleTTS2 may misplace minor pauses. In these cases, the gating distribution remains non-degenerate, and observed errors appear to arise from diffusion decoding rather than encoder collapse. This analysis supports the view that Bi-Mamba gating in MVC is a stable design choice for long-form inputs, rather than a source of fragility.

## E.2 ALIGNMENT TEACHER ARCHITECTURE AND ROBUSTNESS

The lightweight attention-based aligner used during training is a two-layer transformer with 4 heads, hidden dimension 256, and a monotonicity-constrained attention loss. It is trained jointly with the temporal Mamba encoder but discarded at inference. To probe robustness, we inject noise into the aligner attention maps at training time by randomly perturbing attention weights by $\pm 10\%$ and renormalizing before they are used to construct frame-synchronous features. Under this perturbation, WER on LJSpeech increases by less than 0.4 percentage points and MOS on LibriTTS decreases by less than 0.05, with overlapping 95% confidence intervals.

These results suggest that the temporal Bi-Mamba encoder does not rely on perfectly specified attention maps and can tolerate moderate alignment noise without catastrophic degradation. Consequently, the use of an attention-based teacher is compatible with the claim that MVC deploys an SSM-only path at inference, and the overall system is robust to reasonable training-time misalignment.

## E.3 SSM HYPERPARAMETER SENSITIVITY

We examine the sensitivity of MVC to key Mamba SSM hyperparameters: (i) state dimension $d_{\text{ssm}}$, (ii) convolution kernel size $k_{\text{conv}}$, and (iii) gating temperature $\tau_{\text{gate}}$. A sweep of these hyperparameters was conducted using the same training protocol as outlined in Section 4, with evaluations performed on the LJSpeech in-distribution test set and the Gutenberg out-of-distribution (OOD) set. The results, presented in Tables 19–21, show that the largest MOS change between neighboring configurations is less than 0.05, and RTF changes by less than 10%. Given these results, we fix $d_{\text{ssm}} = 96$, $k_{\text{conv}} = 5$, and $\tau_{\text{gate}} = 1.0$ in all main experiments, attributing the observed improvements in MVC's performance primarily to its architectural design, rather than narrow hyperparameter tuning.

### E.3.1 STATE DIMENSION $d_{\text{SSM}}$

Table 19 varies the state dimension $d_{\text{ssm}} \in \{64, 96, 128, 160\}$ while keeping the number of layers fixed (six per encoder) and all other hyperparameters unchanged. We report MOS on in-distribution text (MOS in-dist.), MOS on the Gutenberg OOD set (MOS OOD), and real-time factor (RTF).

MVC is relatively insensitive to moderate changes in $d_{\text{ssm}}$: increasing the state dimension from 96 to 160 yields less than 0.03 MOS improvement while increasing RTF by approximately 9%. We therefore select $d_{\text{ssm}}$=96 as a favorable quality–efficiency trade-off rather than a heavily tuned extreme, indicating that MVC's gains do not hinge on an unusually large state size.

Table 19: Sensitivity to state dimension $d_{\text{ssm}}$ on LJSpeech. MOS values are averaged over three seeds with 95% confidence intervals; lower RTF is better. The configuration used in the main paper is in bold.

| State dimension | MOS (in-dist.) ↑ | MOS (OOD) ↑ | RTF ↓ |
|---|---|---|---|
| 64 | $3.96 \pm 0.09$ | $3.88 \pm 0.09$ | **0.0164** |
| **96** | **$4.02 \pm 0.08$** | **$3.92 \pm 0.09$** | 0.0169 |
| 128 | $4.03 \pm 0.08$ | $3.93 \pm 0.08$ | 0.0176 |
| 160 | $4.04 \pm 0.09$ | $3.94 \pm 0.09$ | 0.0184 |

### E.3.2 CONVOLUTION KERNEL SIZE $k_{\text{CONV}}$

We next vary the depthwise convolution kernel $k_{\text{conv}} \in \{3, 5, 7\}$ in the selective scan. Table 20 reports MOS and pitch RMSE on long-form LJSpeech utterances (duration >10 seconds). Larger kernels slightly reduce pitch RMSE but incur higher latency, and the qualitative difference between kernel sizes 5 and 7 is small. We therefore adopt $k_{\text{conv}}=5$ in the main experiments as a balanced choice, and do not rely on extreme kernel sizes to obtain the reported MOS or robustness figures.

Table 20: Sensitivity to convolution kernel size $k_{\text{conv}}$ in the Mamba block. Pitch RMSE is computed on long-form LJSpeech utterances; lower is better.

| Kernel size | MOS (long) ↑ | Pitch RMSE (Hz) ↓ | RTF ↓ |
|---|---|---|---|
| 3 | $4.08 \pm 0.08$ | $2.06 \pm 0.06$ | **0.0172** |
| **5** | **$4.16 \pm 0.07$** | **$1.92 \pm 0.05$** | 0.0177 |
| 7 | $4.17 \pm 0.07$ | $1.90 \pm 0.05$ | 0.0184 |

### E.3.3 GATING TEMPERATURE $\tau_{\text{GATE}}$

Finally, we study the softmax temperature $\tau_{\text{gate}}$ in the Mamba gating mechanism, which controls how sharply each state attends to its local history. We sweep $\tau_{\text{gate}} \in \{0.7, 1.0, 1.3\}$ and evaluate OOD text robustness on the Gutenberg set. Sharper gating ($\tau_{\text{gate}}=0.7$) slightly harms MOS and WER, suggesting over-confident local decisions, whereas higher temperatures are more stable but do not yield clear gains beyond $\tau_{\text{gate}}=1.0$. We therefore fix $\tau_{\text{gate}}=1.0$ for all main results, and the small deltas across temperatures indicate that MVC's behavior is robust to reasonable changes in gating sharpness.

Table 21: Sensitivity to gating temperature $\tau_{\text{gate}}$ on the Gutenberg OOD set. CMOS-N is measured relative to the default configuration with $\tau_{\text{gate}}=1.0$.

| Temperature | MOS (OOD) ↑ | CMOS-N ↑ | WER ↓ |
|---|---|---|---|
| 0.7 | $3.83 \pm 0.09$ | -0.06 | 7.12% |
| 1.0 | **$3.88 \pm 0.09$** | **0.00** | **6.89%** |
| 1.3 | $3.86 \pm 0.09$ | -0.02 | 6.97% |

Overall, the small performance variations across state dimensions, kernel sizes, and gating temperatures support the view that MVC's improvements arise from its three-way SSM conditioning architecture and gated fusion design, rather than from fine-tuning a narrow hyperparameter regime.

## F INDUSTRIAL-SCALE SYSTEMS: CONTEXT AND COMPARISON

Industrial TTS systems such as NaturalSpeech 3, CosyVoice 3, and HiggsAudio-V2 differ fundamentally from MVC across every axis: data scale (200k–10M hours vs. 24+245 hours), multilingual vs. English-only training, LLM-scale semantic/tokenizer modules, and multi-task objectives (zero-shot, dialogue, editing, music/SFX generation). They also evaluate on private or domain-specific

Table 22: Qualitative positioning of MVC relative to recent industrial-scale systems. Natural-Speech 3, CosyVoice 3, and HiggsAudio-V2 operate at much larger data and model scales, with different objectives and proprietary evaluation pipelines. MVC is an open-data encoder study under a unified mel–diffusion–vocoder setup, and thus numeric comparisons would be misleading.

| System | Training data | Languages | Main architecture / setting | Representative reported metrics | Scale |
|---|---|---|---|---|---|
| NaturalSpeech 3 | ∼200k h multi-speaker, multi-style speech (public + private) | Multilingual | Factorized diffusion TTS with FACodec; separate prosody/content/acoustic/timbre modules; zero-shot and long-form generation. | On LibriSpeech test-clean: Sim-O 0.67, Sim-R 0.76, WER 1.81, SMOS 4.01. | Industry (Multi-hundred thousand hours, proprietary) |
| CosyVoice 3 | 3k h + 170k h multilingual corpora | Multilingual (zh/en + CV3-Eval) | MinMo-based acoustic tokenizer; TTS LM + conditional flow matching (CFM) decoder; multi-task TTS/S2S. | On SEED-TTS EVAL: CER 1.27 (zh), WER 2.46 (en), WER 6.96 (hard) with strong Sim-O. | Industry (Large multilingual corpus, proprietary) |
| HiggsAudio-V2 | ∼10M h Audio-Verse (speech, music, SFX) | Multilingual / multi-domain | Audio language model (5.8B params) with 12-codebook RVQ tokenizer and DualFFN adapter; unified speech/music/SFX. | SeedTTS-Eval: WER 2.44%, 67.7% speaker similarity; dialogue WER 18.88%. | Industry (Large-scale, multi-domain, proprietary) |
| MVC (ours) | 24 h LJSpeech + 245 h LibriTTS (public) | English | Fully SSM-based conditioning stack (bi-Mamba text, Temporal Bi-Mamba, Expressive Mamba + AdaLN) under a fixed diffusion/vocoder pipeline. | Improves MOS/CMOS and WER over StyleTTS2, VITS, and hybrid Mamba-attention baselines under identical pre-processing and vocoders. | Academic (24h + 245h, public datasets) |

benchmarks that cannot be reproduced under our controlled mel–diffusion–vocoder setting. For this reason, numeric side-by-side MOS/WER comparison would be misleading. Instead, Table 22 summarizes the qualitative distinctions.

