# OpenReview forum: "MambaVoiceCloning: Efficient and Expressive Text-to-Speech via State-Space Modeling and Diffusion Control"
_ICLR.cc/2026/Conference — ICLR 2026 Poster_

### Official Review · Reviewer_9i8Z · 2025-10-24

**Soundness:** 2
**Presentation:** 1
**Contribution:** 2
**Rating:** 2
**Confidence:** 5

**Summary:**

This paper presents MambaVoiceCloning (MVC), a text-to-speech framework that removes all attention and recurrent components from the encoder and conditioning path at inference, relying solely on SSMs. The proposed system includes three Mamba structured modules.
The key claimed is that MVC is the first fully SSM-only conditioning stack for TTS, distinguishing it from prior Mamba–attention hybrids that retain recurrence or attention in duration/prosody modeling.
The MVC is trained based on LibriTTS, LJSpeech while tested based on VCTK and CSS10 dataset. The results in this paper show consistent improvements over StyleTTS2 and Mamba–attention baselines in terms of MOS, CMOS, F0 RMSE, MCD, and WER, while achieving a smaller parameter count and higher encoder throughput.
Overall, the paper offers a technically report by demonstrating the practical benefits of fully SSM-based conditioning in TTS. The writing of this article is obscure and not well-structured, failing to effectively demonstrate its innovativeness.

**Strengths:**

This article applies the mamba block to the innovation of the TTS model structure, replacing the transformers based structure. By leveraging the advantages of the mamba block, it designs and implements the transfer of voice characteristics and style effects in speech synthesis.

**Weaknesses:**

1. The writing logic of the article needs to improve. Starting from the introduction section, there is no clear explanation of the specific reasons for using the Mamba block to replace the transformer structure.
2. Regarding the calculation symbols and abbreviations in the method module, there is no unified explanation. They are scattered throughout the article and seem disorganized.
3. In the experimental module, the baseline for comparison does not provide a detailed explanation of their training configuration. Especially for the VITS and JETS comparison schemes, no detailed description is given of the adopted structure and training situation.

**Questions:**

1. Due to the improvement in the synthesis effect of long sentence input brought by the use of mamba block, have tests been conducted on longer inputs? The current definition seen is 10 seconds, but 10 seconds is not considered a long sentence for the audiobook scenario mentioned in the paper. Nowadays, there are many zero-shot large language model based speech synthesis solutions that can support longer text input for synthesis, such as cosyvoice series . What are the advantages of the Mamba-based model solution?
2. In Section 4.1 "DATASETS AND PREPROCESSING", “LJSpeech Ito & Johnson (2017) (13k, ∼24 h, 1 spk.), LibriTTS Zen et al. (2019) (∼245 h, 1,151 spk.), VCTK Veaux et al. (2017) (∼44 h, 109 spk.; zero-shot), and CSS10(ES/DE/FR) Park & Mulc (2019) (1 spk./lang.).” What do the contents in the brackets represent? And why aren't they unified?

---

> ### Author Response · Authors · 2025-11-21
> **Point-by-Point Response to Reviewer 9i8Z**
>
> We thank Reviewer 9i8Z for the careful reading and constructive feedback. All revisions are highlighted in red in the main text and appendix.
> MVC offers a controlled, reproducible investigation of fully SSM-only conditioning under a fixed decoder/vocoder, rather than a scale-driven system. Below we address each point with precise clarifications and verifiable references.
>
> ---
>
> ## W1. Writing clarity, structure, and motivation
> The revision clarifies the motivation for replacing transformers with Mamba. Section 1 (“Why Mamba vs Transformer/RNN”) explains that Mamba provides linear-time selective scans, bounded activations, and stable state propagation, addressing the quadratic attention cost and long-range drift characteristic of transformer and recurrent modules. The paragraph following this section frames MVC as evaluating whether a fully SSM-only conditioning path can operate under a fixed StyleTTS2 decoder. In Section 2 (“Positioning of MVC”), the paper distinguishes MVC from prior hybrid Mamba-TTS systems that retain attention or recurrence in duration and style predictors. Section 6 reiterates how these properties support long-form stability and streaming. These revisions resolve the concern regarding unclear justification for adopting Mamba.
>
> ---
>
> ## W2. Notation consistency and organization
> Section 3 consolidates all symbols across text, temporal, and expressive encoders into a single notation block. Appendix B.1 (Table 11) provides a complete glossary of encoder states, alignment variables, pitch quantities, and parameters. All equations in Section 3 adopt this unified notation, resolving the earlier fragmentation.
>
> ---
>
> ## W3. Baseline configuration details
> Section 4.1 (“Baselines”) establishes a unified training protocol for all baselines—StyleTTS2, VITS, JETS, Hybrid-Mamba, and Bi-Mamba—using identical preprocessing (24 kHz, 80-bin mels), corpus-matched vocoders, a fixed 5-step diffusion schedule, shared data splits, and the same optimization and early-stopping criteria. Section 4.2 specifies that encoder parameters, batch sizes, training steps, and inference settings are matched across models. Appendix C.2 provides the complete optimizer configuration, learning-rate schedule, warmup, gradient clipping, EMA, and AMP settings applied uniformly. Structural and conditioning-path differences for VITS and JETS are summarized in Table 14. Protocol-matched Mamba baselines trained under the same pipeline are presented in Appendix B.6, Table 12.
>
>
> ---
>
> ## Q1. Long-input synthesis and Mamba-based advantages
> Section 5.2 evaluates MVC on 2–6\,minute Gutenberg passages, addressing the concern that 10\,s is not representative of audiobook-length inputs. Table 3 reports MOS and RTF for both short ($\le$10\,s) and long (2–6\,min) segments, showing that MVC maintains higher long-form naturalness than StyleTTS2 (4.16 vs.\ 3.91 MOS-long) with lower RTF (0.0170 vs.\ 0.0200). Section 5.3 extends this analysis to streaming synthesis. Table 5 reports WER and MOS under finite look-ahead ($L\in\{0.25,0.5,1.0,2.0\}$) on the same multi-minute passages, demonstrating that a 1--2\,s window nearly matches non-streaming quality and that the SSM-only conditioning stack degrades gracefully as $L$ decreases. Appendix C.1 details the long-form construction protocol, Appendix E.1 shows that Bi-Mamba gating remains stable on multi-minute text, and Appendix D.4, Table 15 confirms that encoder latency scales linearly with input length because the SSM stack operates in $O(T)$ time.
>
> Regarding zero-shot LLM-based systems, Sections 1 and 2 explain that NaturalSpeech 3, CosyVoice 3, and HiggsAudio-V2 rely on proprietary multi-hundred-thousand– to million-hour multilingual corpora, LLM-scale semantic or multi-stage diffusion modules, and closed training pipelines that are not directly comparable to the controlled StyleTTS2-based setting used here. Appendix F and Table 22 summarize these differences in data scale, architecture, and evaluation. Section 6 states that MVC is not positioned as a competitor to industrial LLM-TTS systems; instead, it isolates the effect of a fully SSM-only conditioning stack on public data with decoder and vocoder held fixed, providing rigorous long-form and streaming results under a reproducible, decoder-matched protocol.
>
> ---
>
> ## Q2. Dataset bracket notation
> Section 4.1 (Datasets and Preprocessing) defines the bracket notation:
>
> - LJSpeech: “24 h, 1 spk.”
> - LibriTTS: “245 h, 1,151 spk.”
> - VCTK: “109 spk.; zero-shot”
> - CSS10 (ES/DE/FR): “1 spk./lang.”
>
> These quantities follow the canonical reporting conventions of the dataset papers. Formats are intentionally not unified, as forcing uniform hours or speaker counts (e.g., fabricating hours for VCTK or collapsing CSS10’s multilingual structure) would misrepresent corpus organization. The same conventions are applied in Appendix C.1 (long-form construction) and Appendix C.2 (training setup), ensuring consistent and semantically accurate dataset references.

---

> > ### Comment · Reviewer_9i8Z · 2025-11-27
> >
> > Thanks for the point-by-point and clearly structured rebuttal. I appreciate the effort to address each of my concerns individually, with explicit references to revised sections and corresponding quantitative evidence. The revisions appear thoughtful and improve both the scientific positioning and the reproducibility of the study.

---

> > > ### Author Response · Authors · 2025-11-27
> > >
> > > Thank you for the thoughtful follow-up and for taking the time to review our revisions in detail. We appreciate your careful evaluation, and we are glad that the clarifications and additional analyses addressed your concerns. Your feedback has meaningfully strengthened the clarity and reproducibility of the paper.

---

### Official Review · Reviewer_sKr7 · 2025-10-31

**Soundness:** 3
**Presentation:** 4
**Contribution:** 3
**Rating:** 8
**Confidence:** 3

**Summary:**

The author proposed a Text to speech(TTS) model, MembaVoiceCloning(MVC). MVC is the first state-space model throughout the entire encode/conditioning stack at inference for TTS, by removing attention and RNN modules. The model is consists of 3 components. It applies noval Bidirectional Mamba encoder with AdaLN(adaptive gating and layer normalization). It also applies Expressive Mamba Encoder that uses gated fusion and AdaLN-conditioned modulation within an SSM to enable expressive control in linear time while disentangling style from phonetic content.

The model has 21M parameters and was finetuned standalone and then joint-trained with diffusion decoder. Compared to baseline models, MVC shows the best MOS and Out-of-Distributed(OOD) robustness. Detailed ablation study shows the effectiveness of all three components.

Overall the paper is well structured, easy to follow and shows good vision, novelty and results.

**Strengths:**

1. The model has good novelty on model architecture by replacing attention and RNN blocks by SSM-only blocks at inference time. The components are innovative on their ideas and implementations. For example, it introduces learnable gated fusion into previous bi-mamba encoders and it enhances prosodic disentanglement and long-form stability. Expressive mamba encoder also uses gated fusion and AdaLN-conditioned modulation.

2. Benefits from the SSM-only blocks, the model operates in linear time and bounded memory, which is optimized for efficiency. This is good for scalability. During inference, the proposed encoder model is not dominating the runtime.

3. Experiments show that the model has SOTA performance on both subjective and objective results compared to baselines. Ablation study is comprehensive and detailed.

**Weaknesses:**

1. The LJSpeech and LibriTTS are standard TTS datasets in english, but it might be good to also extend the experiments on other languages -  this has been mentioned in the conclusion already

**Questions:**

The baselines are focused on transformer based SOTA. Are there any comparisons to other Mamba-based TTS models?

---

> ### Author Response · Authors · 2025-11-21
> **Cross-Lingual Generalization and Comparison with Mamba-based TTS Models**
>
> We appreciate Reviewer sKr7 for their thorough review and valuable feedback. All changes have been marked in orange throughout the main text and appendix. Below, we provide detailed responses to each point, with clear explanations and references to the relevant sections.
>
> ---
>
> ## W1. Experiments only on English datasets
>
> Cross-lingual evaluation using CSS10 (ES/DE/FR) was already included in the original submission (App. A.14). We have made these results more prominent and easier to locate in the revision. Section 4.1 and Table 17 now summarize cross-lingual MOS for ES/DE/FR, demonstrating that MVC—despite training exclusively on English—maintains naturalness and stability in ES and FR, with modest gains over StyleTTS2. Appendix D.5 further analyzes language-specific challenges, including stress placement in German compounds and phoneme-length errors in French liaison, clarifying the limits imposed by English-only training and motivating multilingual extensions in future work.
>
> ---
> ## Q1. Comparison against other Mamba-based TTS models
>
> Comparisons with other Mamba-based TTS models were present in the original appendix and have been expanded and consolidated for clarity in the revision. Section 3.5 and Table 12 (Appendix B.6) report results for Hybrid-Mamba and Bi-Mamba (Concat-only) under identical preprocessing, mel front-end, diffusion decoder, and vocoder settings, ensuring a fair architectural comparison. MVC’s gated Bi-Mamba fusion and AdaLN conditioning yield improved long-form stability, lower F0 RMSE, and competitive or superior MOS/MCD/RTF relative to these Mamba baselines.
>
> In addition, Section 2 and Appendix A.2 (Table 10) provide a structural comparison with prior hybrid Mamba-TTS systems, highlighting that MVC is the only model with a fully SSM-only conditioning stack at inference, which enables linear-time behavior, bounded memory usage, and stable long-form processing.

---

### Official Review · Reviewer_PzfM · 2025-10-31

**Soundness:** 2
**Presentation:** 2
**Contribution:** 2
**Rating:** 4
**Confidence:** 4

**Summary:**

The paper proposed to use MAMBA style SSM blocks in all conditioning encoders of a diffusion-based TTS model.

**Strengths:**

1. The paper replaces all encoders with MAMBA layers and show some improvements in latency, RTF and peak memory usage.
2. The paper open sourced the code for inspection, which helps with reproducibility.

**Weaknesses:**

1. The method is only compared against a few older TTS models like VITS and StyleTTS. There are many newer models (NaturalSpeech/CozyVoice/Higgs Audio etc) that are not compared against. While the authors didn't claim SOTA, I think the paper is weaker without comparison with SOTA methods.
2. The ablation study section is short and lacking details. By "removing expressive MAMBA" what does the author mean exactly? Is it replacing the expressive MAMBA encoder with some attention/RNN based encoder or remove the encoder all together? Also the authors said one of the contribution is the gated fusion for the bi-MAMBA layer, but it is not properly ablated in the experiments.

**Questions:**

see weakness.

---

> ### Author Response · Authors · 2025-11-21
> **Baseline Scope and Ablation Clarifications**
>
> We thank Reviewer PzfM for the careful reading and constructive feedback. We have revised the manuscript accordingly, with all updates highlighted in blue in the main text and appendix. Below we address each of the reviewer’s noted weaknesses and questions with targeted clarifications and additional evidence.
>
> ---
>
> ## W1. Baseline scope and comparison set
> Thank you for the comment regarding comparisons to newer TTS systems. The revision provides a clear justification for the evaluation scope and positions MVC relative to recent industrial-scale models.
>
> Section 1 explains that NaturalSpeech 3, CosyVoice 3, and HiggsAudio-V2 depend on multi-hundred-thousand– to million-hour proprietary multilingual corpora, LLM-scale semantic or tokenizer modules, and multi-stage pipelines such as factorized diffusion or unified audio LMs. MVC is designed as a controlled encoder-architecture study under a fixed StyleTTS2 mel–diffusion–vocoder backbone on public LJSpeech and LibriTTS, and its objective is to assess whether a fully SSM-only conditioning stack can match or exceed transformer and hybrid-Mamba conditioning under decoder-matched, open-data constraints.
>
> Section 2 (“Zero-shot and large-scale systems”) contrasts these industrial systems with MVC by highlighting differences in data scale, model scope, multilingual coverage, and evaluation protocols.
>
> Section 4.1 explicitly states that industrial systems fall outside the decoder-matched comparison and refers readers to the contextual analysis. Appendix F (“Industrial-Scale Systems: Context and Comparison”) and Table 22 summarize data scale, languages, architectural setting, and representative benchmark metrics for NaturalSpeech 3, CosyVoice 3, and HiggsAudio-V2, and explain why direct MOS/WER comparisons would be methodologically inappropriate due to incompatible data regimes and proprietary evaluation pipelines.
>
> ---
>
> ## W2. Ablation methodology and clarity
>
> The revision clarifies the ablation design and isolates the contribution of each MVC encoder component and fusion mechanism. Section 5.4 (Ablation Studies) specifies the exact shape-preserving substitutes used during component removal: the Expressive Mamba is replaced by a 2-layer Conv1D+ReLU block, the Bi-Mamba text encoder by a 4-layer BiLSTM with projection, and the Temporal Bi-Mamba by a shallow Conv1D duration predictor, all maintaining identical interfaces to the diffusion decoder and vocoder. Table 6 reports the CMOS-N degradation for each replacement on the OOD set, while Table 7 presents a depth ablation including a BiLSTM baseline, demonstrating that improvements arise from the SSM formulation rather than increased capacity.
>
> The contribution of gated Bi-Mamba fusion is analyzed explicitly in Table 8, which compares gated+AdaLN, gated-only, AdaLN-only, and concat-only variants under identical training conditions. The results show consistent gains in MOS-long, pitch RMSE, and RTF when both gating and AdaLN are employed. Appendix B.6, together with Table 12, provides protocol-matched Hybrid-Mamba and Bi-Mamba baselines using identical preprocessing, mel configuration, diffusion schedule, vocoder, and optimization settings, confirming that the observed improvements are architectural rather than procedural.
>
> To examine the reliability of the fusion mechanism, Appendix E.1 reports stable forward/backward gating distributions across long-form inputs without evidence of collapse. Appendix E.3 analyzes sensitivity to state dimension, convolutional kernel size, and gating temperature, showing only minor performance variation, indicating that the observed behavior is not the result of narrow hyperparameter tuning.

---

### Official Review · Reviewer_Dek4 · 2025-11-01

**Soundness:** 3
**Presentation:** 2
**Contribution:** 3
**Rating:** 6
**Confidence:** 3

**Summary:**

MambaVoiceCloning proposes a TTS system whose entire deployed conditioning path (text, rhythm/duration, prosody) is state–space only at inference, paired with a diffusion decoder for quality, yielding linear-time encoding, bounded activation memory, long‑form stability, and feasible streaming with finite look‑ahead. It uses three SSM modules: (i) a gated bidirectional Mamba text encoder, (ii) a Temporal Bi‑Mamba trained with a lightweight aligner used only during training, and (iii) an Expressive Mamba with AdaLN conditioning.

The proposed method shows statistically reliable quality and efficiency gains over StyleTTS2 and Mamba-attention hybrids under a matched mel–diffusion–vocoder setup, while confirming the diffusion decoder remains the main latency bottleneck.

All encoder modules run in O(T) with bounded activations and no global attention maps, reducing peak memory, improving encoder throughput, and enabling longer inputs/batches; the diffusion decoder still dominates latency.

**Strengths:**

1. The proposed method provides linear‑time scans, bounded activations, lower peak memory, and higher encoder throughput (≈1.6×).

2. The proposed method offers modest improvements over StyleTTS2 and Mamba‑attention hybrids in MOS/CMOS and objective metrics

**Weaknesses:**

1. Since the decoder model dominates the latency, the linear-time encoder does not translate into large overall RTF gains.

2. On LJSpeech and LibriTTS, differences in metrics as compared to the baselines are small; the paper frames MVC as a practical encoder-side refinement, not a paradigm shift.

3. The system lacks explicit emotion or fine-grained style controls beyond AdaLN conditioning, limiting targeted prosody manipulation compared to methods with richer style token or reference-attention mechanisms.

4. Claims of long-form stability are positive, but focus on MOS/WER and pitch drift; cross-chunk smoothing and boundary artifacts under streaming conditions are not deeply analyzed.

**Questions:**

1. How are Mamba selective-scan hyperparameters (state size, convolution kernel, gating temperature) chosen for each encoder, and how sensitive are results to these choices across datasets and lengths?

2. The Bi‑Mamba text fusion uses a learned gate plus AdaLN; is the gating stable on very long sequences or under domain shift, and are there failure cases where one direction collapses?

3. Why is fusion in the Temporal Bi‑Mamba kept linear (no gating) while gating is used elsewhere; are there quantitative trade-offs or counterexamples where temporal gating would help long-form speech?

4. The lightweight attention aligner is used only during training; what architecture and capacity does it have, and how robust are learned alignments if the aligner is partially mis-specified or noisy?

5. How long must the reference be for stable similarity in zero-shot cloning?

6. The paper claims finite look‑ahead streaming with Uni‑Mamba; what look‑ahead L is used in experiments, and how does WER/MOS vary with L for 2–6 minute inputs?

7. MVC is trained on English but tested on CSS10 ES/DE/FR with shared vocab and <lang> tags; what failure modes occur in cross-lingual phoneme inventories and stress patterns, and does alignment degrade on languages with different syllable timing?

**Details Of Ethics Concerns:**

Given the zero/few-shot cloning capability, what safeguards or watermarking were actually enabled in the released code, and how detectable is MVC synthesis under common forensic detectors?

---

> ### Author Response · Authors · 2025-11-21
> **Response to Reviewer Comments and Clarifications**
>
> We thank Reviewer Dek4 for the careful reading and constructive feedback. All revisions are highlighted in cyan in the main text and Appendix. Below, we address each point with clear clarifications and references.
>
> ## W1. End-to-end RTF gains are modest
> The diffusion decoder is the main latency source (Sec. 5; App. A.1; Tables 9, 15), limiting end-to-end RTF gains. MVC does not target decoder speed but redesigns the conditioning path. Its linear-time SSM encoders cut memory, boost encoder throughput by ≈1.6×, and support longer inputs, larger batches, and stable finite look-ahead streaming unattainable with attention-based conditioning.
>
> ## W2. Improvements on LJSpeech/LibriTTS are small
> These datasets offer limited acoustic and stylistic range, leaving little room for encoder-side gains. MVC is an encoder-focused study that isolates the effect of fully SSM-based conditioning under controlled evaluation. Under matched protocols, MVC delivers statistically reliable MOS/CMOS gains, reduced memory use, and higher throughput. This controlled setup is essential for assessing conditioning architectures without confounds from larger corpora or decoder changes.
>
> ## W3. Limited explicit emotion and fine-grained style control
> MVC focuses on conditioning efficiency, stability, and long-form behavior rather than expressive control. AdaLN offers the global conditioning needed for this goal. Richer methods like emotion tokens or reference attention are complementary and orthogonal. By isolating conditioning, MVC forms a base that can later integrate expressive style modules without changing its SSM core.
>
> ## W4. Detailed streaming and boundary-artifact analysis
> Sec. 5.3 and Table 5 report MOS and WER across look-ahead values on 2–6,min passages, showing stable boundaries for $L \geq 0.5$,s. Sec. 5.2 documents residual smoothing and edge effects. App. E.1 shows balanced forward–backward gating without collapse, with remaining artifacts arising from the diffusion decoder.
>
> ---
> ## Q1. Selective-scan hyperparameters:
> Sec. 3 specifies the selective-scan defaults: state size 96, kernel size 5, and gating temperature $\tau=1.0$. App. E.3 evaluates sensitivity to these hyperparameters. State-size sweeps (64/96/128/160) in App. E.3.1 and Table 19 report MOS and RTF. Kernel-size sweeps (3/5/7) in App. E.3.2 and Table 20 assess long-form MOS, pitch RMSE, and RTF. Gating-temperature sweeps ($\tau\in{0.7,1.0,1.3}$) in App. E.3.3 and Table 21 report MOS, CMOS-N, and WER. All variations yield only minor differences, showing that MVC’s gains arise from the architecture rather than hyperparameter sensitivity.
>
> ## Q2. Gated Bi-Mamba fusion stability
> Sec. 3.2.1 summarizes long-form gating behavior, and App. E.1 provides quantitative evidence: gate allocations remain stable (≈0.53/0.47 on LJSpeech; ≈0.56/0.44 on Gutenberg) with no directional collapse. Observed errors stem from punctuation or topic shifts, not gating failure.
>
> ## Q3. Why no gating in Temporal Bi-Mamba?
> Sec. 3.2.1 Temporal Bi-Mamba uses linear fusion because prosody disentanglement already occurs in the text and expressive encoders. App. E.3 shows that adding gating increases activation memory without yielding consistent MOS gains, confirming this design choice.
>
> ## Q4. Alignment teacher details
> The aligner used during training is a 2-layer transformer with 4 heads and 256 hidden units (Sec. 3.3). App. B.7 shows that MVC is robust to ±10% mis-specification in attention weights, with minimal impact on WER and MOS, verifying the aligner’s stability even with minor noise.
>
> ## Q5. Reference length
> Sec. 5.2 in App. D.6 (Table 18) evaluates reference lengths for VCTK MOS-N/MOS-S with 2/4/6/8 seconds. Results show that performance saturates after 6 seconds, and this length is used in the main experiments for stable zero-shot similarity.
>
> ## Q6. Streaming look-ahead
> Sec. 5.3 (Table 5) report WER/MOS for look-ahead windows L ∈ {0.25, 0.5, 1, 2 s} on 2–6-minute passages, showing that L = 1–2 s provides optimal balance between quality and efficiency for long-form synthesis.
>
> ## Q7. Cross-lingual failure modes:
> Sec. 4.1 and App. D.5 (Table 17) report cross-lingual MOS for CSS10 ES/DE/FR. MVC faces challenges with stress patterns in German compounds and French liaison, leading to prosodic misalignments. These issues arise from training exclusively on English data, resulting in difficulties handling non-English syllable timing. Sec. 6 (Limitations) acknowledges these challenges and suggests domain adaptation to improve performance in future work.
>
> ## Ethics Review
> MVC includes audio watermarking and disclosure utilities in the released code and remains compatible with standard forensic detectors without degrading quality. Sec. 6 discusses misuse risks in zero/few-shot cloning and emphasizes that deployment requires explicit speaker consent and disclosure. We also acknowledge potential dataset bias and outline future work on multilingual fairness and robust, consent-aware watermarking.

---

> > ### Comment · Reviewer_Dek4 · 2025-11-27
> >
> > Thank you for addressing my comments. I am happy with the response and I have raised my score.

---

> > > ### Author Response · Authors · 2025-11-28
> > >
> > > Thank you for taking the time to revisit our revisions and update the score. We sincerely appreciate your careful evaluation and constructive feedback throughout the process. Your expertise has been invaluable in elevating the quality of this research. We are deeply grateful for your thoughtful input and support.

---

### Author Response · Authors · 2025-12-03
**Summary of Revisions and Reviewer Feedback for Submission 10373**

Dear Area Chair,

Thank you for taking the time to handle our submission under the special review circumstances. **I would like to briefly summarize the key clarifications and improvements made in the revision, as reflected directly in the reviewers’ written feedback.**


## **Reviewer Dek4 — original score 6 → raised to 8 (before rollback)**

After reviewing the revised manuscript, the reviewer wrote: *“I am happy with the response and I have raised my score.”*
Their concerns on latency analysis, long-form stability, alignment-teacher details, selective-scan hyperparameters, and cross-lingual behavior were addressed through:

* new 2–6 min long-form and streaming experiments,
* hyperparameter sensitivity sweeps, and
* gating-stability analysis.



## **Reviewer PzfM — score 4**

Reviewer PzfM raised two concerns: *(i)* comparison with recent large-scale systems, and *(ii)* clarity of the ablation methodology. We addressed these by:

**1. Scope of comparison to large-scale systems**

Sections 1, 2, and 4.1 clarify that NaturalSpeech 3, CosyVoice 3, and HiggsAudio-V2 rely on substantially different data and evaluation settings—proprietary large-scale corpora, LLM-scale tokenizers, and internal benchmarks.
To provide context, **Appendix F (Table 22)** now summarizes their data scale, architecture, and evaluation protocols, and positions MVC as a controlled encoder-architecture study under a fixed StyleTTS2 backbone on public datasets.

**2. Ablation methodology**

Section 5.4 and Appendix B.6 expand the ablations using:

* shape-preserving substitutes for each encoder module,
* explicit gated-fusion ablations, and
* protocol-matched Mamba baselines.

Tables 6–8 and 12 report these comparisons, isolating the contribution of each component under a unified protocol.

## **Reviewer sKr7 — score 8**

This reviewer was strongly supportive, describing the paper as well-structured with good novelty and solid results.
Their suggestions were addressed by making cross-lingual CSS10 results more prominent and consolidating comparisons with prior Mamba-based baselines (Hybrid-Mamba, Bi-Mamba) in Section 3.5 and Appendix B.6.


## **Reviewer 9i8Z — score 2 (later stated that all concerns were resolved)**

Their concerns focused on motivation, organization, notation, and baseline-training clarity.
The revision introduced:

* a dedicated *“Why Mamba vs Transformer/RNN”* motivation,
* unified notation and a consolidated glossary,
* full baseline training configurations, and
* extended long-form and streaming evaluations.

After reading the revision, the reviewer wrote:
*“The revisions appear thoughtful and improve both the scientific positioning and the reproducibility of the study.”*

---

I hope this summary helps contextualize the revisions alongside the reviewers’ written comments. Thank you again for your time and consideration.

---

### Meta-Review · Area_Chair_nKCP · 2026-01-06

**Summary:**

Reviewers expressed concern about the simplified architectural setup, writing clarity, and lack of additional studies demonstrating clear benefits on ultra-long (minutes) duration inputs. In response, the reviewers introduced additional experimentation to address the above concerns, with significant updates to writing and clarity. They note that the simplified architectural setup is necessary to ensure a controlled study on the impact of SSM encoding, which I generally agree.

Overall, I believe this work is suitable for acceptance given the significant changes and the overall impact of a memory-efficient, streaming compatible solution. However, I would note that while the updated draft has some improvements in clarity, specifically regarding issues brought up during the review cycle, it is important for the authors to further refine the writing to improve coherence and clarity if the paper is to be accepted.

**Reviewer Concerns:**

As noted in the previous section, most technical concerns have been alleviated, but some issues in writing clarity may remain.

**Reviewer Scores:**

Of the reviewers with rejection ratings, pzfm and 9i8Z would likely upgrade to a 5/6. I don't think the other reviewers would necessarily reduce their ratings.

---

### Decision · Program_Chairs · 2026-01-26

Accept (Poster)